# Spatiotemporal resolution of germinal center Tfh cell differentiation and divergence from central memory CD4+ T cell fate

Fangming Zhu [1,8], Ryan J. McMonigle[1,8], Andrew R. Schroeder[1], Xianyou Xia[1], David Figge[2], Braxton D. Greer[3], Edahí González-Avalos[4], Diego O. Sialer [1], Yin-Hu Wang [1], Kelly M. Chandler[1], Adam J. Getzler[5], Emily R. Brown[1], Changchun Xiao [6], Olaf Kutsch[3], Yohsuke Harada [7], Matthew E. Pipkin [5] & Hui Hu [1]✉

Follicular helper T (Tfh) cells are essential for germinal center (GC) B cell responses. However, it is not clear which PD-1+CXCR5+Bcl6+CD4+ T cells will differentiate into PD-1hiCXCR5hiBcl6hi GC-Tfh cells and how GC-Tfh cell differentiation is regulated. Here, we report that the sustained Tigit expression in PD-1+CXCR5+CD4+ T cells marks the precursor Tfh (pre-Tfh) to GC-Tfh transition, whereas Tigit−PD-1+CXCR5+CD4+ T cells upregulate IL-7Rα to become CXCR5+CD4+ T memory cells with or without CCR7. We demonstrate that pre-Tfh cells undergo substantial further differentiation at the transcriptome and chromatin accessibility levels to become GC-Tfh cells. The transcription factor c-Maf appears critical in governing the pre-Tfh to GC-Tfh transition, and we identify Plekho1 as a stage-specific downstream factor regulating the GC-Tfh competitive fitness. In summary, our work identifies an important marker and regulatory mechanism of PD-1+CXCR5+CD4+ T cells during their developmental choice between memory T cell fate and GC-Tfh cell differentiation.

T follicular helper (Tfh) cells are a unique CD4+ T cell subset that provides help to B cells and is essential for germinal center (GC) formation[1,2]. It has been proposed that Tfh cell differentiation is a multi-stage process involving PD-1+CXCR5+ pre-Tfh and PD-1hiCXCR5hi GC-Tfh cell steps[2–5], with transcription factor Bcl6 playing a central role[6–8]. Studies have shown that dendritic cells are sufficient to prime naïve CD4+ T cells to generate PD-1+CXCR5+Bcl6+ cells[9], which then require interactions with B cells to become PD-1hiCXCR5hiBcl6hi GC-Tfh cells[4,9–11]. Although it has been suggested that not all the activated PD-1+CXCR5+CD4+ T cells will enter B cell follicles and become GC-Tfh cells, and there are follicular helper-like central memory CD4+ T cells

expressing CXCR5 and CCR7[12,13], it is not known how to distinguish which PD-1+CXCR5+CD4+ T cells will enter B cell follicles and become GC-Tfh cells while others have a different fate[14].

Tigit was identified as a surface protein that is expressed on various T lymphocytes and exerts immunosuppressive effects[15]. Later, Tigit was found to play important roles in NK cells and anti-tumor immunity[16]. As an important immunomodulator, Tigit is induced in activated CD4+ T cells and some CXCR5+CD4+ T cells[15,17]. However, the Tigit expression pattern and its role in Tfh cell differentiation are poorly understood.

In Tfh studies, the use of cell surface staining of CXCR5 to resolve CXCR5hi cells from CXCR5+ cells has varied considerably[6,18–21]. Although

[1]Department of Microbiology, School of Medicine, University of Alabama at Birmingham, Birmingham, AL 35294, USA. [2]Department of Pathology, School of Medicine, University of Alabama at Birmingham, Birmingham, AL 35294, USA. [3]Department of Medicine, School of Medicine, University of Alabama at Birmingham, Birmingham, AL 35294, USA. [4]Division of Signaling and Gene Expression, La Jolla Institute for Immunology, La Jolla, CA 92037, USA. [5]Department of Immunology and Microbiology, The Scripps Research Institute, Jupiter, FL 33458, USA. [6]Department of Immunology and Microbiology, The Scripps Research Institute, La Jolla, CA 92037, USA. [7]Faculty of Pharmaceutical Sciences, Tokyo, University of Science, Chiba 278-8510, Japan. [8]These authors contributed equally: Fangming Zhu, Ryan J. McMonigle. ✉e-mail: huihu@uab.edu

transcriptome and chromatin accessibility studies have been carried out to understand the Tfh cell differentiation, the majority of the studies compare CXCR5⁻ non-Tfh cells to bulk CXCR5⁺ Tfh cells and few have involved Bcl6hi cells[6,19,22–27], and little is known about the stage-specific regulation of PD-1hiCXCR5hiBcl6hi GC-Tfh cell differentiation.

The transcription factor c-Maf has broad functions in CD4⁺ T cell subsets[28]. In addition to its identified role in human Tfh cell differentiation in vitro[29], a previous study reported that c-Maf is expressed early after T cell activation and is important for Tfh cell differentiation in vivo as well[30]. More interestingly, a recent study on Thpok-mediated regulation of Tfh cell differentiation showed that co-expression of Bcl6 and c-Maf, but not Bcl6 alone, in Thpok-deficient CD4⁺ T cells was able to rescue the generation of later stage CXCR5⁺PD-1hi Tfh cells[25], suggesting that c-Maf likely plays a critical role in GC-Tfh cell differentiation. How c-Maf may specifically regulate GC-Tfh cell differentiation is not known.

Here we report that the sustained Tigit expression in PD-1⁺CXCR5⁺CD4⁺ T cells marks the pre-Tfh to GC-Tfh transition and c-Maf is a critical regulator in this transition. Furthermore, we find that the c-Maf downstream factor Plekho1 plays an important role in regulating the competitive fitness of GC-Tfh cells. Our findings distinguish two subsets of PD-1⁺CXCR5⁺CD4⁺ T cells that have divergent fates of GC-Tfh cell differentiation versus formation of CXCR5⁺CD4⁺ T cell memory with or without the central memory phenotype and reveal a stage-specific regulatory mechanism in GC-Tfh cell differentiation.

## Results

### Sustained Tigit expression in PD-1⁺CXCR5⁺CD4⁺ T cells is associated with GC-Tfh cell differentiation

In C57BL/6 mice infected with influenza virus A/Puerto Rico/8/34 (PR8), we found that the GC B cell response in the draining mediastinal lymph node (medLN) started after day 7 p.i., peaked around day 14 p.i., and then reduced (Supplementary Fig. 1a, b). The kinetics of the influenza nucleocapsid protein (NP₃₁₁₋₃₂₅)-tetramer⁺ (NP⁺) CD44hiPD-1hiCXCR5hi GC-Tfh cell response were consistent with those of the GC B cell response (Supplementary Fig. 1c, d). Interestingly, by using a Bcl6-protein reporter mouse line[31], we found that although there was a sizable percentage of NP⁺CD44hiCD4⁺ T cells expressing high levels of PD-1 and CXCR5 at day 6–7 p.i., there was no GC formation yet at this time (Fig. 1a, b). These early-stage PD-1hiCXCR5hiCD4⁺ T cells expressed Bcl6 protein at a level similar to PD-1⁺CXCR5⁺ cells (both were much higher than in naïve CD4⁺ T cells), but far lower than that of PD-1hiCXCR5hi GC-Tfh cells when GCs were fully induced (Fig. 1b, c). The results suggest that the simple PD-1 and CXCR5 staining levels or qualitative Bcl6 staining are not sufficient to indicate the GC-Tfh cell differentiation.

Studies have shown that Tigit is induced in activated CD4⁺ T cells and some CXCR5⁺CD4⁺ T cells[15,17], but the Tigit expression pattern during Tfh cell differentiation is not clear. In PR8 infection, we found that almost all of the NP⁺CD44hiCD4⁺ T cells upregulated Tigit at day 7 p.i. in the medLN (Supplementary Fig. 1e). By day 14 p.i., about half of the NP-specific PD-1⁺CXCR5⁺ cells still expressed Tigit, whereas most of the PD-1hiCXCR5hi GC-Tfh cells were Tigit⁺ (Supplementary Fig. 1f). After day 14 p.i., the high percentage of Tigit⁺ cells in the PD-1hiCXCR5hi GC-Tfh population was maintained, but Tigit expression in the PD-1⁺CXCR5⁻ non-Tfh and PD-1⁺CXCR5⁺ cell populations gradually decreased (Supplementary Fig. 1f). We also examined Tigit expression in an OT-II adoptive transfer/PR8-OVA model, where the PR8 virus has been engineered to express the OVA epitope[32]. SMARTA T cell receptor (TCR) transgenic mice were used as recipients to reduce the competition between donor OT-II and host CD4⁺ T cells[5,33,34]. A similar Tigit expression pattern and kinetics were observed in OT-II cells responding to PR8-OVA (Supplementary Fig. 1g and Fig. 1d, e). Using Fucci2-cell-cycle reporter mice[35], we found that donor OT-II-Fucci2 cells were actively expanding at the early stage of the response, but by day 14 p.i. the proliferation of all donor cell populations was drastically diminished (Supplementary Fig. 1h, i). We also transferred OT-II cells into C57BL/6

mice and found similar Tfh cell differentiation and Tigit expression patterns, but with a reduced percentage of PD-1hiCXCR5hi GC-Tfh cells (Supplementary Fig. 1j, k) that was likely due to competition.

While Bcl6 expression was low in PD-1⁺CXCR5⁻ non-Tfh cells, we found that Bcl6 expression was progressively higher from Tigit⁻ to Tigit⁺ PD-1⁺CXCR5⁺ cells, with the highest Bcl6 expression in GC-Tfh cells (Fig. 1f). Studies have suggested Tigit could be a direct Bcl6 target[36]; we therefore examined Tigit expression in the absence of Bcl6. At day 6 p.i., Tigit expression was induced in activated Bcl6-deficient OT-II cells, although to a lesser extent than wild-type (WT) OT-II cells (Fig. 1g). By day 14 p.i. when Bcl6-deficient OT-II cells failed to develop into Tfh cells, few of them maintained Tigit expression (Fig. 1g), indicating that after Tigit is induced, sustained Tigit expression in PD-1⁺CXCR5⁺CD4⁺ T cells is strongly associated with Tfh cell differentiation and in particular, the differentiation towards GC-Tfh cells.

To further explore this idea, we examined the Tigit expression pattern in donor OT-II cells in situ. By day 6–7 p.i. when there were no GCs (Fig. 1b), although most of the OT-II cells upregulated Tigit (Fig. 1d), they were either in the T cell zone or coming towards the T-B border (Fig. 2a). After GC formation was fully established on day 14 p.i., we found that most of the donor OT-II cells in the GCs were still Tigit⁺ (Fig. 2b), and the donor cells inside the B cell follicles were also Tigit⁺ (Fig. 2b). At this timepoint, whereas some Tigit⁺ OT-II cells were found along the T-B border, the amount of the Tigit⁺ OT-II cells in the T cell zone was drastically reduced (Fig. 2b). Overall, these results were consistent with the kinetics of the Tigit expression pattern observed first in early activated CD4⁺ T cells, then in non-Tfh, PD-1⁺CXCR5⁺CD4⁺ T cells, and GC-Tfh cells (Fig. 1d, e), supporting that the sustained Tigit expression in PD-1⁺CXCR5⁺CD4⁺ T cells seems to mark the further differentiation of such cells into GC-Tfh cells.

In vitro, we found that TCR stimulation induced Tigit expression in CD4⁺ T cells as reported[15,37], but it was quickly down-regulated when TCR stimulation was withdrawn (Fig. 2c); upon re-stimulation with TCR, Tigit expression was recovered (Fig. 2d), suggesting that Tigit expression in activated CD4⁺ T cells can be sustained by continued or repeated TCR stimulation−a setting reminiscent of interactions between T and B cells at the T-B border and inside the GC[3,11,38–40].

To examine the function of Tigit in regulating Tfh cell differentiation and/or Tfh helper function to B cells, we combined an shRNAmir knockdown approach and the OT-II/B1-8i co-transfer immunization model[34,41]. Donor OT-II cells treated with shRNAmir in vitro were co-transferred together with B1-8i transgenic B cells into the Bcl6f/fCD4-CreTg recipient mice followed by immunization with NP-OVA plus LPS. The shRNAmir against Tigit (shTigit) drastically reduced the expression of Tigit on the cell surface (Supplementary Fig. 2a), but the differentiation of PD-1hiCXCR5hi GC-Tfh cells was not affected (Supplementary Fig. 2b). Comparing the shTigit group and the control shCD19 group, we found that there were no obvious differences in the differentiation of GC B cells and formation of plasmablasts (Supplementary Fig. 2c), light zone (LZ)/dark zone (DZ) ratios (Supplementary Fig. 2d), or the IgG1 and IgG2b isotype switch (Supplementary Fig. 2e). These collective results suggest that Tigit signaling is overall not critical for GC-Tfh cell differentiation or Tfh helper function to GC B cells.

### Transcriptome and chromatin accessibility analyses of Tigit⁻ and Tigit⁺ Tfh cells

To investigate the differences between Tigit⁻PD-1⁺CXCR5⁺, Tigit⁺PD-1⁺CXCR5⁺, and PD-1hiCXCR5hi GC-Tfh cells, we sorted naïve OT-II and donor OT-II sub-populations responding to PR8-OVA for transcriptome and chromatin accessibility analyses.

We partitioned differentially expressed genes (DEGs) into five clusters, with Cluster A highlighting a unique set of genes specifically upregulated in PD-1hiCXCR5hi GC-Tfh cells (Fig. 3a). Overall, the differences between Tigit⁻ and Tigit⁺ PD-1⁺CXCR5⁺ cells were relatively small, but there were increased DEGs between Tigit⁻PD-1⁺CXCR5⁺ and

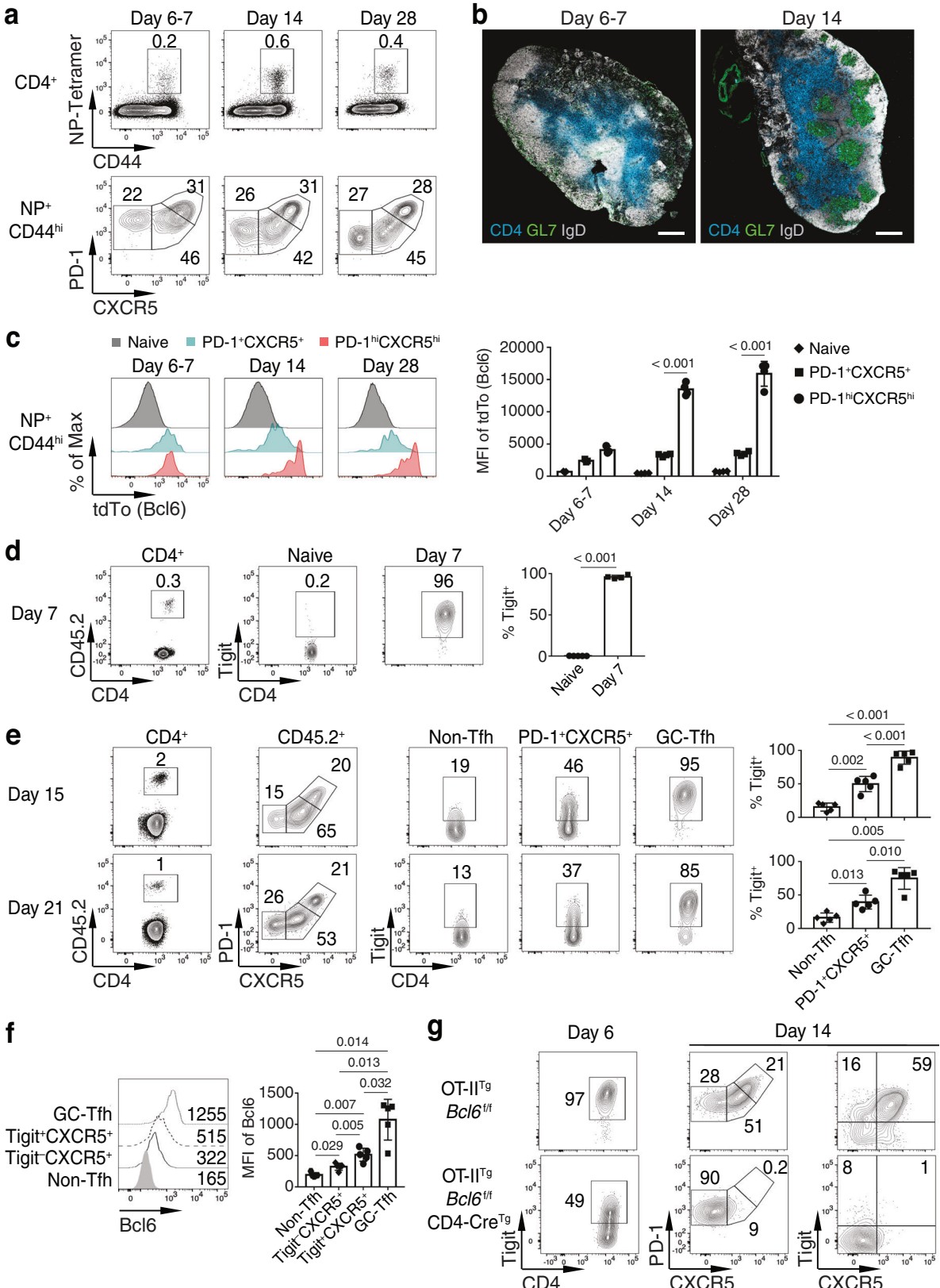

PD-1$^{hi}$CXCR5$^{hi}$ GC-Tfh cells compared to DEGs between Tigit$^+$PD-1$^+$CXCR5$^+$ and PD-1$^{hi}$CXCR5$^{hi}$ GC-Tfh cells (1077 vs. 499 genes, respectively; Fig. 3b and Supplementary Fig. 3a). The expression levels of select non-Tfh/Tfh hallmark genes examined by RT-PCR demonstrated that Tigit$^+$PD-1$^+$CXCR5$^+$ cells have a gene expression pattern in between Tigit$^-$PD-1$^+$CXCR5$^+$ and PD-1$^{hi}$CXCR5$^{hi}$ GC-Tfh cells (Supplementary

Fig. 3b). Ingenuity Pathway Analysis (IPA) found Tigit$^+$PD-1$^+$CXCR5$^+$ cells were enriched for genes involved in mTOR, HIF-1α, and ICOS-ICOSL signaling pathways compared to Tigit$^-$PD-1$^+$CXCR5$^+$ cells (Fig. 3c), despite the similar expression levels of ICOS (Supplementary Fig. 3b), suggesting a likely engagement of Tigit$^+$PD-1$^+$CXCR5$^+$ cells with B cells. The ICOS-ICOSL, Ephrin receptor, and HIF-1α signaling pathways were

**Fig. 1 | Sustained Tigit expression in CXCR5⁺CD4⁺ T cells is associated with GC-Tfh cell differentiation. a–c** Bcl6-protein reporter mice were intranasally infected with PR8. **a** At days 6–7, 14, and 28 p.i., the NP⁺CD44ʰⁱCD4⁺ T cells in the medLN were analyzed for PD-1 and CXCR5 staining. **b** Confocal microscopy analysis of expression of CD4 (blue), GL7 (green), and IgD (white) in the medLN was performed at days 6–7 and 14 p.i. Scale bars, 400 μm. **c** Indicated populations of NP⁺CD44ʰⁱCD4⁺ T cells at days 6–7, 14, and 28 p.i. were analyzed for tdTomato (tdTo, Bcl6-reporter) expression (days 6–7, $n = 4$; day 14, $n = 4$; day 28, $n = 4$). **d–f** Purified OT-II cells were transferred into CD45.1⁺ SMARTA recipient mice followed by intranasal infection with PR8-OVA. (**d**) Naïve OT-II and donor OT-II at day 7 p.i. and (**e**) indicated donor cell populations at days 15 and 21 p.i. in the medLN were

analyzed for Tigit staining (day 7, $n = 4$; day 15, $n = 5$; day 21, $n = 5$). **f** Indicated donor cell populations at day 21 p.i. in the medLN were analyzed for Bcl6 intracellular staining ($n = 5$). **g** Purified OT-II cells from OT-IIᵀᵍBcl6^f/f or OT-IIᵀᵍBcl6^f/fCD4-Creᵀᵍ mice were transferred into CD45.1⁺ SMARTA recipient mice followed by intranasal infection with PR8-OVA. Indicated donor cell populations in the medLN were analyzed for Tigit, PD-1, and CXCR5 staining at days 6 and 14 p.i. Data in **a–g** are representative (or pooled) results of at least two independent experiments. Bars represent average ±SD. The *P*-values were determined by a two-tailed unpaired *t*-test (**d**), a one-way ANOVA with Tukey's multiple comparisons test (**e**, **f**), or a two-way ANOVA with Sidak's multiple comparisons test (**c**). Source data are provided as a Source Data file.

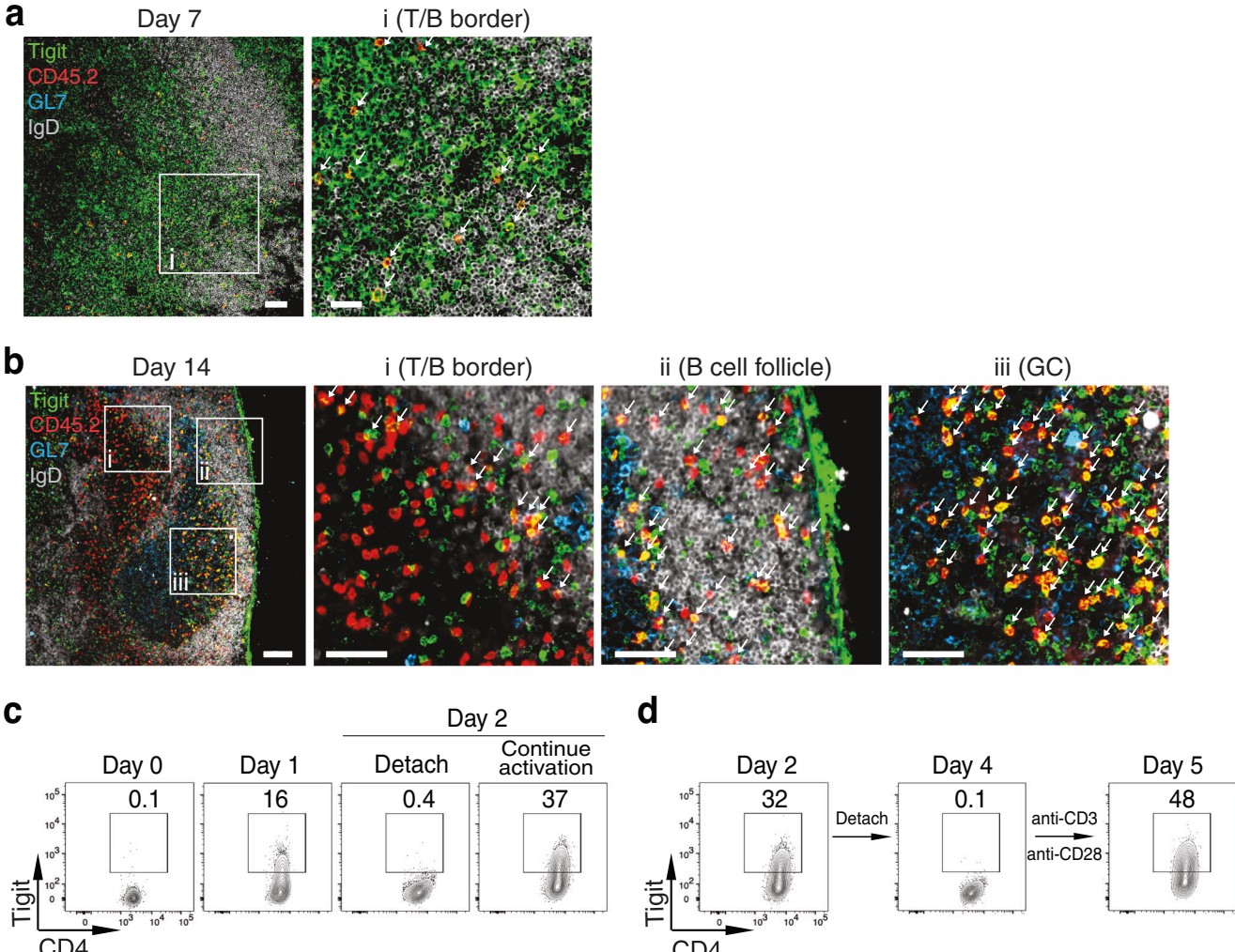

**Fig. 2 | Tigit⁺CD4⁺ T cell localization and regulation of Tigit expression in activated CD4⁺ T cells. a, b** Purified CD45.2⁺ OT-II cells were transferred into CD45.1⁺ C57BL/6 recipient mice followed by intranasal infection with PR8-OVA. Confocal microscopy analysis of expression of Tigit (green), CD45.2 (red), GL7 (blue), and IgD (white) in the medLN was performed at days 7 (**a**) and 14 (**b**) p.i. Insets of the T/B border (i), B cell follicle (ii), and GC (iii), as indicated, are shown at higher magnification, with each arrow (pointing to yellow) representing a cell stained with both Tigit and CD45.2. Scale bars, 50 μm and 25 μm (for zoom-in views) in **a**, and 100 μm and 50 μm (for zoom-in views) in **b**. **c, d** Purified CD4⁺ T cells

from C57BL/6 mice were activated in vitro with anti-CD3/CD28 antibodies. Tigit staining was analyzed in: **c** naïve cells (Day 0), day 1 activated cells (Day 1), day 2 activated cells (Continue activation), and cells stimulated for 1 day, followed by transfer of cells to a new well for another day without further stimulation (Detach), and **d** CD4⁺ T cells activated for 2 days (Day 2) and expanded in culture for 2 days (Day 4), then re-stimulated with anti-CD3/CD28 antibodies overnight (Day 5). Data in **a–d** are representative results of at least two independent experiments. Source data are provided as a Source Data file.

further enriched in PD-1ʰⁱCXCR5ʰⁱ GC-Tfh cells, and genes involved in ERK/MAPK, NF-κB, 14-3-3, CD28 co-stimulation, and glycolysis pathways seemed to be specifically increased in PD-1ʰⁱCXCR5ʰⁱ GC-Tfh cells (Fig. 3c). PTEN signaling, reported to be negatively regulated by Bcl6[42], was reduced in PD-1ʰⁱCXCR5ʰⁱ GC-Tfh cells (Fig. 3c).

ATAC-seq analysis showed that similarly, there were more differences in chromatin accessibility between PD-1ʰⁱCXCR5ʰⁱ GC-Tfh and Tigit⁺PD-1⁺CXCR5⁺ cells than between Tigit⁺ and Tigit⁻ PD-1⁺CXCR5⁺ cells (Fig. 3d and Supplementary Fig. 3c, d). Differentially accessible regions (DARs) were partitioned into 6 clusters and transcription factor (TF)

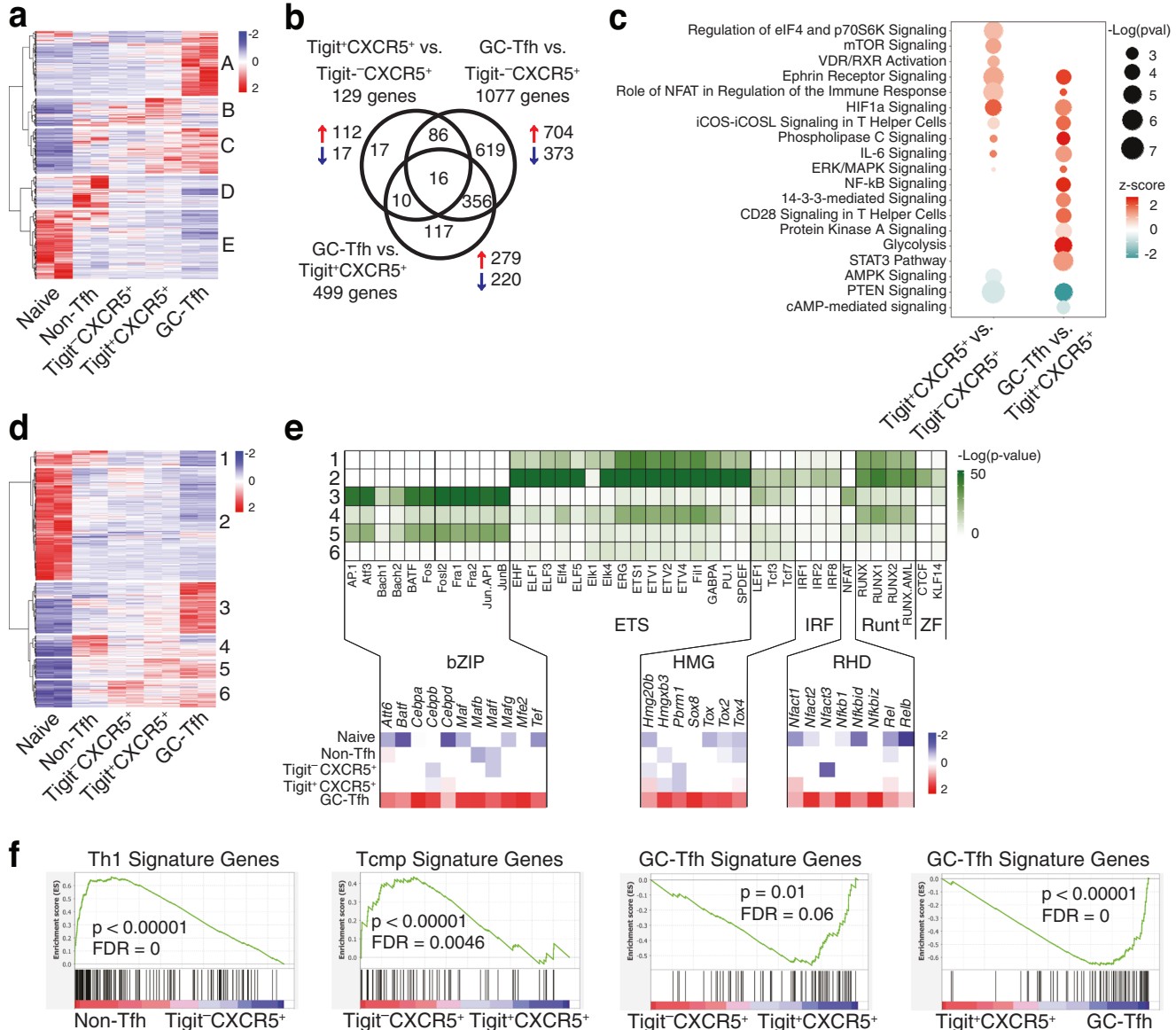

**Fig. 3 | Transcriptome and chromatin accessibility analyses in Tfh cell differentiation.** Purified OT-II cells were transferred into CD45.1+ SMARTA recipient mice followed by intranasal infection with PR8-OVA. **a**–**c** Donor OT-II cells at day 21 p.i. in the medLN were analyzed by RNA-seq. **a** Clustered heatmap of naïve and indicated donor cell populations. **b** Venn diagram of DEGs and **c** IPA for the indicated comparisons. **d, e** Donor OT-II cells at day 14 p.i. in the medLN were analyzed by ATAC-seq. **d** Clustered heatmap of naïve and indicated donor cell populations. **e** Transcription factor binding motif enrichment in ATAC-seq peak clusters from Fig. 3d (top), with RNA-seq gene expression of selected transcription factor family members (bottom). **f** GSEA of indicated donor cell populations using published Th1, Tcmp, and GC-Tfh signature gene sets. RNA-seq and ATAC-seq samples are independent biological replicates. Source data are provided as a Source Data file.

binding motif analysis was performed (Fig. 3d, e). Approximately 30% of DARs were associated with promoter regions (Supplementary Fig. 3e). The chromatin regions with increased accessibility in PD-1hiCXCR5hi GC-Tfh cells (Cluster 3) were associated with bZIP, HMG, and RHD TF binding motifs (Fig. 3e). Expression of several TFs from the bZIP, HMG, and RHD TF families—including *Batf* and *Maf*, *Tox* and *Tox2*, and *Nfkb1* and *Nfatc1/2*, respectively (Fig. 3e)—were consistent with their reported roles in Tfh cell differentiation[26,29,30,43–46]. The TF Ascl2 has been suggested to initiate CXCR5 induction and Tfh cell programming[47], although a separate study has also shown that Ascl2 is not detectable when initial CXCR5 expression is induced[48]. In our study, *Ascl2* was found to be expressed mainly in the GC-Tfh cells (Supplementary Fig. 3b), and, consistently, our ATAC-seq analysis showed that the *Ascl2* locus seemed to become accessible only in GC-Tfh cell stage (Supplementary Fig. 3d). Thus, the considerable and unique changes of gene expression and chromatin accessibility between PD-1+CXCR5+ and

PD-1hiCXCR5hi GC-Tfh cells suggest pre-Tfh cells undergo substantial further differentiation to become GC-Tfh cells.

Using published gene sets for Th1 cells, central memory CD4+ T cell precursors (Tcmp), and GC-Tfh cells[13], Gene Set Enrichment Analysis (GSEA) indicated that Tigit+PD-1+CXCR5+ cells were closer to PD-1hiCXCR5hi GC-Tfh cells, whereas Tigit−PD-1+CXCR5+ cells aligned more with a Tcmp cell phenotype (Fig. 3f). These results suggest that within the PD-1+CXCR5+CD4+ T cell population, the PD-1+CXCR5+ cells that sustain the Tigit expression will likely differentiate into GC-Tfh cells, whereas the PD-1+CXCR5+ cells that downregulate the Tigit expression will differentiate towards the direction of memory formation.

## The divergent differentiation of Tigit− and Tigit+ PD-1+CXCR5+CD4+ T cells

To determine the potential divergent differentiation of Tigit+ versus Tigit− PD-1+CXCR5+CD4+ T cells and, in particular, how

Tigit⁻PD-1⁺CXCR5⁺CD4⁺ T cells may be associated with CD4⁺ T cell memory, we examined donor OT-II cells in the medLN at day 28 p.i. (Fig. 4a). PR8-OVA infection induced a robust population of donor PD-1⁺CXCR5⁺CD4⁺ T cells in the spleen as well, although the overall OT-II cell response was stronger in the draining medLN than in the spleen (Supplementary Fig. 4a). The differentiation of PD-1$^{hi}$CXCR5$^{hi}$ GC-Tfh cells was also comparatively weaker in the spleen (Fig. 4b). The Tigit expression pattern in donor OT-II cells in the spleen was similar to that in the medLN (Supplementary Fig. 4b). Interestingly, in both the medLN and the spleen, we found that within the PD-1⁺CXCR5⁺CD4⁺ T cell population, it was mainly the Tigit⁻ cells that upregulated IL-7Rα (Fig. 4a, b). The majority of non-Tfh cells were Tigit⁻ and IL-7Rα⁺; in contrast, Tigit⁺PD-1⁺CXCR5⁺ and PD-1$^{hi}$CXCR5$^{hi}$ GC-Tfh cells were IL-7Rα⁻ (Fig. 4a, b). We also examined Tigit versus IL-7Rα expression pattern in the LCMV envelope glycoprotein (GP) immunization model in which the GC response diminished 5 weeks after the immunization (Supplementary Fig. 5a, b). We found that similarly, more than half of the GP-tetramer⁺ (GP⁺) PD-1⁺CXCR5⁺CD4⁺ T cells downregulated Tigit with some of them starting to express IL-7Rα around day 14 p.i. (Supplementary Fig. 5c). As the response was progressively decreased, more Tigit⁻GP⁺PD-1⁺CXCR5⁺ cells increased IL-7Rα expression (Supplementary Fig. 5c).

To determine the functional differences between Tigit⁺ and Tigit⁻ PD-1⁺CXCR5⁺CD4⁺ T cells and whether the Tigit⁺PD-1⁺CXCR5⁺ cells possess more GC-Tfh cell differentiation characteristics, we infected congenic groups of CD45.1⁺ and CD45.1⁺CD45.2⁺ C57BL/6 mice with PR8. At day 14 p.i., we sorted CD45.1⁺CD45.2⁺CD44⁺Tigit⁺IL-7Rα⁻PD-1⁺CXCR5⁺ and CD45.1⁺CD44⁺Tigit⁻IL-7Rα⁺PD-1⁺CXCR5⁺ CD4⁺ T cells and co-transferred them into CD45.2⁺ recipient mice followed by PR8 infection. We found that by day 14 p.i., whereas the CD45.1⁺ donor cells (from Tigit⁻IL-7Rα⁺PD-1⁺CXCR5⁺ cells) had higher cell recovery (Fig. 4c, d), the CD45.1⁺CD45.2⁺ donor cells (from Tigit⁺IL-7Rα⁻PD-1⁺CXCR5⁺ cells) elicited a much more robust GC-Tfh cell response than did the CD45.1⁺ donor cells with significantly higher GC-Tfh to PD-1⁺CXCR5⁺CD4⁺ T cell ratios (Fig. 4e), suggesting that Tigit⁺PD-1⁺CXCR5⁺CD4⁺ T cells have differentiated further into the GC-Tfh direction than their Tigit⁻ counterparts, confirming the functional divergence in the response.

IL-7Rα⁺ non-Tfh cells expressed CXCR6 and higher levels of Psgl-1 than IL-7Rα⁻PD-1⁺CXCR5⁺CD4⁺ T cells (Fig. 5a, b), consistent with non-Tfh characteristics[49,50]. Most of the IL-7Rα⁺ non-Tfh and IL-7Rα⁻PD-1⁺CXCR5⁺CD4⁺ T cells from the medLN were CD62L⁻ and CCR7⁻, but the splenic IL-7Rα⁻PD-1⁺CXCR5⁺CD4⁺ T cells had a higher percentage expressing CCR7 (Fig. 5c). Intriguingly, when OT-II cells were transferred into wild-type C57BL/6 recipient mice (with increased competition compared to SMARTA recipient mice) followed by PR8-OVA infection, a higher percentage of IL-7Rα⁺ non-Tfh and IL-7Rα⁻PD-1⁺CXCR5⁺CD4⁺ T donor cells expressed CCR7 in the draining medLN (Fig. 5d), suggesting that a relatively weak CD4⁺ T cell response, either in a non-draining lymphoid organ (e.g. spleen) or in a competitive environment, may lead to a central memory phenotype during memory formation.

In our PR8 infection model, the IL-7Rα⁺PD-1⁺CXCR5⁺CD4⁺ T cells always had higher percentages of CCR7⁺ cells than those of IL-7Rα⁺ non-Tfh cells in both the draining lymph node and the spleen (Fig. 5c) consistent with a previous study using a *Listeria monocytogenes* bacterial infection model[12]. Interestingly, in the GP protein immunization model, we found that the IL-7Rα⁺PD-1⁺CXCR5⁺CD4⁺ T cells and IL-7Rα⁺ non-Tfh cells had similar percentages of CCR7⁺ and CCR7⁺CD62L⁺ cells (Fig. 5e). Collectively, all these results suggest that the subset of Tigit⁻ cells within the PD-1⁺CXCR5⁺CD4⁺ T cell population become the memory precursor/memory cells. Whether the CXCR5⁻ and CXCR5⁺ CD4⁺ T memory cells express central memory phenotype seems to be determined by factors that include the type of infection/antigen challenge, the magnitude of the response induced in draining or non-draining secondary lymphoid organs, and the competition within the local environment.

## Stage-specific regulation of GC-Tfh cell differentiation

To explore the mechanism of pre- to GC-Tfh cell differentiation, we cross-examined Cluster A genes that were specifically expressed in PD-1$^{hi}$CXCR5$^{hi}$ GC-Tfh cells and Cluster 3 gene loci with increased chromatin accessibility in PD-1$^{hi}$CXCR5$^{hi}$ GC-Tfh cells (Fig. 6a). Cluster A was divided into two groups of genes: genes with (List I) and genes without (List II) a PD-1$^{hi}$CXCR5$^{hi}$ GC-Tfh specific increase in chromatin accessibility (Fig. 6a). Using IPA, it was striking that many List I genes were enriched in the signaling pathways enhanced in PD-1$^{hi}$CXCR5$^{hi}$ GC-Tfh cells (Fig. 6b).

Among List I genes (Fig. 6c), c-Maf has been shown to be important for Tfh cell differentiation[25,29,30]; the underlying mechanism, however, is incompletely understood. Our RNA-seq and ATAC-seq analyses showed that specifically from PD-1⁺CXCR5⁺CD4⁺ T cells to PD-1$^{hi}$CXCR5$^{hi}$ GC-Tfh cells, c-Maf had both increased mRNA expression as well as chromatin accessibility (Fig. 6d, e). We performed intracellular staining and confirmed that c-Maf did have increased protein levels in PD-1$^{hi}$CXCR5$^{hi}$ GC-Tfh cells (Fig. 6f). Subsequently, in the PR8 infection model, we found that whereas the c-Maf deletion by CRISPR/Cas9 (c-Maf-CRISPR) in naïve CD4⁺ T cells negatively affected the overall CXCR5⁺ Tfh cell development (with increased non-Tfh cell percentages) (Fig. 6g), still a robust population of CXCR5⁺CD4⁺ T cells was generated at the peak of the CD4⁺ T cell response (Fig. 6g). More importantly, however, there seemed to be a specific blockade of PD-1$^{hi}$CXCR5$^{hi}$ GC-Tfh cell development after the generation of PD-1⁺CXCR5⁺CD4⁺ T cells (Fig. 6g), suggesting that c-Maf likely plays a critical role in regulating the pre- to GC-Tfh cell transition.

To further test this idea, we cross-examined a published c-Maf ChIP-seq dataset[51] and our List I and II genes to search for potential c-Maf targets that may play stage-specific roles in regulating GC-Tfh cell differentiation. We found that the majority of List I and II genes—especially the List I genes—were potential c-Maf targets (Supplementary Fig. 6a, b). The pleckstrin homology domain-containing protein 1 (Plekho1, also known as CKIP-1) has been shown to interact with various types of proteins and regulate multiple signaling pathways including the PI3K/Akt pathway[52–55]. Our RNA-seq and ATAC-seq results showed that Plekho1 had significantly increased mRNA expression levels and chromatin accessibility in PD-1$^{hi}$CXCR5$^{hi}$ GC-Tfh cells (Fig. 7a, b). In in vitro activated CD4⁺ T cells, we found that retroviral over-expression of c-Maf (Supplementary Fig. 6c) significantly increased the mRNA expression levels of Plekho1, particularly under the Tfh cell culture condition (Fig. 7c).

To determine the function of Plekho1 in Tfh cell differentiation, we transferred donor OT-II cells treated in vitro with shRNAmir against Plekho1 (sh*Plekho1*) together with the OT-II cells treated with control shRNAmir against CD19 (sh*CD19*) into CD45.1⁺ SMARTA recipient mice followed by RR8-OVA infection. In this competitive environment, whereas sh*Plekho1* did not seem to affect the overall non-Tfh versus PD-1⁺CXCR5⁺CD4⁺ T cell development, an effect was observed between the PD-1⁺CXCR5⁺ and PD-1$^{hi}$CXCR5$^{hi}$ compartments: the further differentiation of PD-1⁺CXCR5⁺ pre-Tfh cells to PD-1$^{hi}$CXCR5$^{hi}$ GC-Tfh cells was significantly impaired (Fig. 7d). These results were confirmed by the CRISPR/Cas9-mediated deletion of Plekho1 (Plekho1-CRISPR) in naïve OT-II cells in the PR8-OVA infection model (Supplementary Fig. 7a), suggesting that Plekho1 does not affect early phase CXCR5⁺CD4⁺ T cell development but rather plays a later, stage-specific role in regulating the PD-1$^{hi}$CXCR5$^{hi}$ GC-Tfh cell differentiation. To examine how Plekho1-deficient GC-Tfh cells may regulate GC B cell responses, we transferred Plekho1-CRISPR naïve OT-II cells into the *Bcl6*$^{f/f}$CD4-Cre$^{Tg}$ recipient mice followed by PR8-OVA infection. We found that in a non-competing environment with no control wild-type GC-Tfh cells, Plekho1-CRISPR GC-Tfh cell differentiation appeared to be normal, and they also supported a normal GC B cell differentiation, isotype switch and LZ/DZ distribution (Supplementary Fig. 7b–e). Thus, our results suggest that the increased expression of Plekho1 by

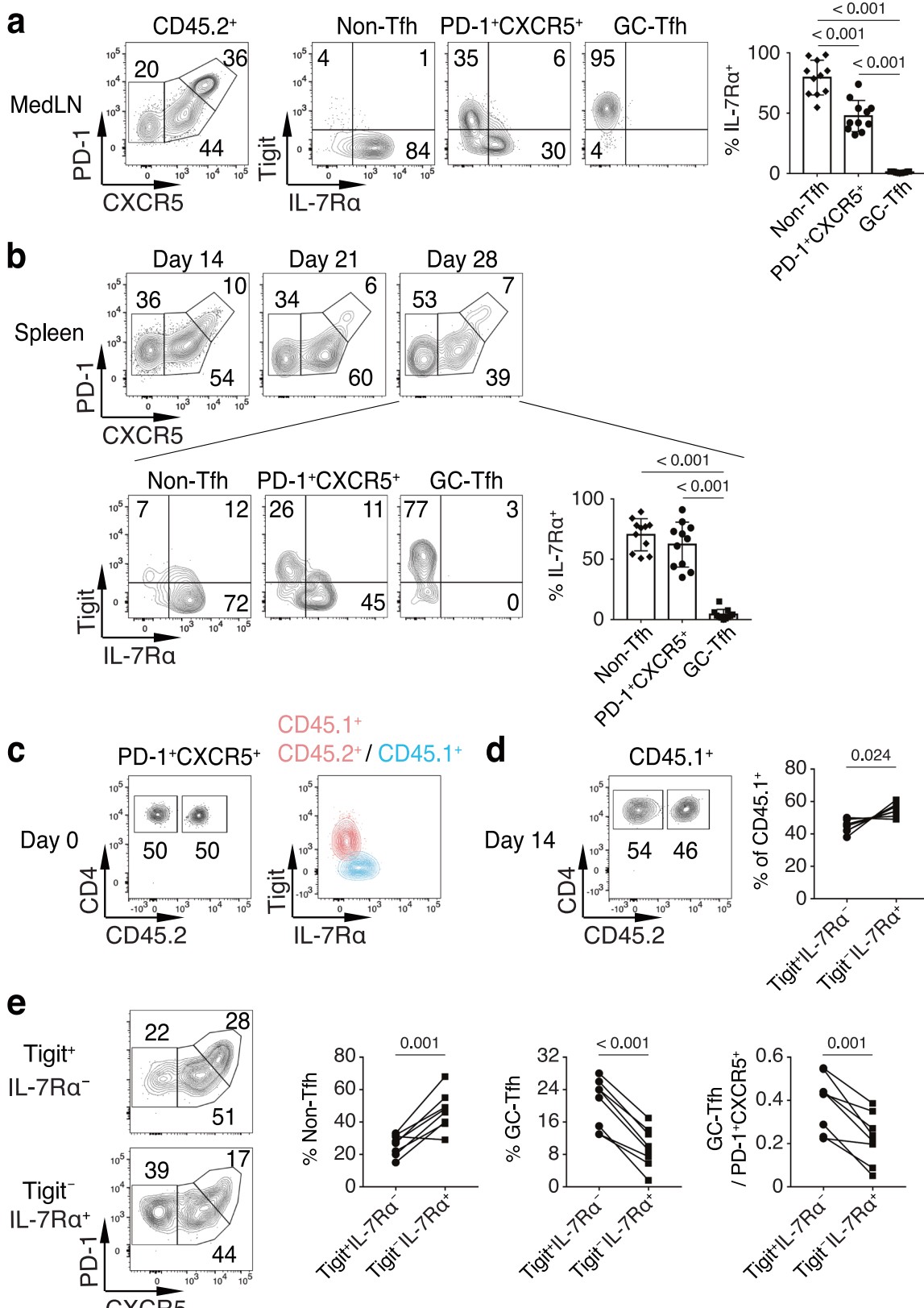

c-Maf during pre- to GC-Tfh transition is critical for the competitive fitness of GC-Tfh cells.

To understand the mechanism underlying Plekho1-mediated regulation of the competitive fitness of GC-Tfh cells, we sorted sh*CD19* and sh*Plekho1* GC-Tfh cells from the competitive environment and performed RNA-seq. Compared to control sh*CD19* GC-Tfh cells, we

found 271 genes decreased and 361 genes increased in sh*Plekho1* GC-Tfh cells (Fig. 7e), with changes involved in multiple pathways (Supplementary Fig. 8a). In particular, we noted the changed expression levels of many Foxo1 targets[56] (Fig. 7f and Supplementary Fig. 8b), indicating increased Foxo1 activity in sh*Plekho1* GC-Tfh cells. Consistently, using Enrichr analysis of transcription factors[57], we found

**Fig. 4 | The divergent differentiation of Tigit⁻ and Tigit⁺ PD-1⁺CXCR5⁺CD4⁺ T cells. a, b** Purified OT-II cells were transferred into CD45.1⁺ SMARTA recipient mice followed by intranasal infection with PR8-OVA. Indicated donor cell populations at day 28 p.i. in the medLN (**a**) or spleen (**b**) were analyzed for PD-1, CXCR5, Tigit, and IL-7Rα staining (n = 11). **c–e** CD45.1⁺CD45.2⁺ and CD45.1⁺ C57BL/6 mice were intranasally infected with PR8. CD45.1⁺CD45.2⁺CD44⁺Tigit⁺IL-7Rα⁻PD-1⁺CXCR5⁺ (Tigit⁺IL-7Rα⁻) and CD45.1⁺CD44⁺Tigit⁻IL-7Rα⁺PD-1⁺CXCR5⁺ (Tigit⁻IL-7Rα⁺) CD4⁺ T cells at day 14 p.i. were sorted and co-transferred into CD45.2⁺ C57BL/6 recipient mice followed by intranasal infection with PR8. **c** Ratio of the two donor cell populations (left panel) and Tigit and IL-7Rα expression on indicated donor cell populations (right panel) before transfer. **d** Ratio of the two donor cell populations (left panel) and quantification of relative percentages at day 14 p.i. (right panel, n = 8). **e** Donor CD4⁺ T cells at day 14 p.i. in the medLN were analyzed for PD-1 and CXCR5 staining, with quantification of indicated populations and PD-1ʰⁱCXCR5ʰⁱ GC-Tfh to PD-1⁺CXCR5⁺ cell ratios (n = 8). Data in **a–e** are representative (or pooled) results of at least two independent experiments. Bars represent average ±SD. The P-values were determined by a two-tailed paired t-test (**d, e**), or a one-way ANOVA with Tukey's multiple comparisons test (**a, b**). Source data are provided as a Source Data file.

that the Foxo1 pathway was greatly enriched in sh*Plekho1* GC-Tfh cells (Fig. 7g). An early study has shown that a mutant form of Foxo1, the Foxo1^AAA, which enforces Foxo1 nuclear localization, reduces the GC-Tfh cell differentiation[58]. Indeed, using ImageStream[59], we found that compared to control sh*CD19* GC-Tfh cells, although the total Foxo1 amounts were the same (Fig. 7h), more Foxo1 was retained in the nucleus in sh*Plekho1* GC-Tfh cells (Fig. 7h). Taken together, our study suggests that in pre- to GC-Tfh cell transition, the increased c-Maf promotes Plekho1 expression, which in turn activates Foxo1 and induces its translocation out of nucleus, and this c-Maf/Plekho1/Foxo1 axis plays a stage-specific role in regulating the competitive fitness of GC-Tfh cells.

## Discussion

After PD-1⁺CXCR5⁺CD4⁺ T cells are generated in the early stage of an immune response, the different potential fates of these cells are important for our understanding of the CD4⁺ T cell response. Our study has revealed that after the early extensive cell proliferation phase is passed, the continued Tigit expression in PD-1⁺CXCR5⁺CD4⁺ T cells is strongly associated with PD-1ʰⁱCXCR5ʰⁱ GC-Tfh cell differentiation. In contrast, it is mainly the Tigit⁻ subset of PD-1⁺CXCR5⁺CD4⁺ T cells that upregulate IL-7Rα to give rise to CXCR5⁺CD4⁺ memory T cells. Using Tigit as a marker, our findings have clarified the issue of PD-1⁺CXCR5⁺CD4⁺ T cells being precursors for both GC-Tfh cells and CCR7⁺/⁻CXCR5⁺CD4⁺ memory T cells.

Regarding cell surface staining of Tfh cells, our study of the Bcl6-protein reporter mice has shown the limitations of using only PD-1/CXCR5 staining or qualitative positive versus negative Bcl6 intracellular staining to define GC-Tfh cells. The findings will resolve some of the confusion regarding of the kinetics of GC-Tfh cell differentiation in many infection or immunization models.

In CD4⁺ T cell responses to *Listeria monocytogenes* and lymphocytic choriomeningitis virus (LCMV) infection, studies have shown that CCR7 is mainly re-expressed in CXCR5⁺ rather than CXCR5⁻ CD4⁺ T memory cells[12,13], leading to the designation of CXCR5⁺CD4⁺ T cells as the central memory T cells[12,13]. This phenomenon was also observed in the CD4⁺ T cell response to influenza virus infection in our study. However, by comparing the CD4⁺ T cell responses in different recipient mice with or without competition, we also demonstrate that CCR7 can also be re-expressed in CXCR5⁻ CD4⁺ T memory cells, and consistently, we observed enhanced CCR7 expression in the spleen where the overall CD4⁺ T cell response is weaker than in the draining lymph node. Furthermore, with the same i.n. route of immunization, there was a higher percentage of CCR7⁺CD62L⁺CD4⁺ T memory cells recovered in the protein immunization model than in the influenza viral infection model, and, notably, equal percentages of CCR7⁺ and CCR7⁺CD62L⁺ cells within the CXCR5⁻ versus CXCR5⁺ CD4⁺ T memory cells. Collectively, our study suggests that the strength of the T cell response—influenced by the competition within the local environment and the location in the draining or non-draining secondary lymphoid organs—and the type of infection/antigen challenge seem to play a critical role in determining whether CXCR5⁻ or CXCR5⁺ CD4⁺ T memory cells carry a central memory phenotype.

In GC-Tfh cell generation, the substantial transcriptome and chromatin accessibility differences between Tigit⁺PD-1⁺CXCR5⁺CD4⁺ T cells and PD-1ʰⁱCXCR5ʰⁱ GC-Tfh cells suggest that after the PD-1⁺CXCR5⁺CD4⁺ T cell phase, the further pre- to GC-Tfh cell development is not due to the simple commitment or maintenance of this population but rather represents a complex further differentiation process. We have provided evidence suggesting that c-Maf exerts an important function in regulating the pre- to GC-Tfh transition. Consistently, c-Maf expression was increased in GC-Tfh cells compared to PD-1⁺CXCR5⁺CD4⁺ T cells. At the PD-1⁺CXCR5⁺CD4⁺ T cell stage, transcriptome analyses indicated that Tigit⁺ cells were one step closer to the GC-Tfh differentiation stage than their Tigit⁻ counterparts, and notably, c-Maf mRNA and protein had already begun to increase in Tigit⁺PD-1⁺CXCR5⁺CD4⁺ T cells. Studies have shown that TCR stimulation increases c-Maf expression[60]. Considering that c-Maf has also been implicated in a positive feed forward loop of self-regulation[61], it is very likely that the repeated T-B interactions during the GC response increase c-Maf expression to high levels; meanwhile, c-Maf controls additional downstream factors and reinforces the commitment to the GC-Tfh differentiation program.

We have identified a c-Maf downstream factor, Plekho1, which, via regulating the Foxo1 pathway, exerts a stage-specific regulation of the competitive fitness of GC-Tfh cells. This stage-specific nature of action supports that pre- to GC-Tfh cell differentiation has its unique programming and regulation, though at present the more intricate changes at the chromatin and gene expression levels and the characteristics of GC-Tfh cells are still not fully understood. Studies have reported that when the GC response fades, PD-1ʰⁱCXCR5ʰⁱ GC-Tfh cells downregulate PD-1, CXCR5, and Bcl6 expression levels[20,22]. Whereas how to distinguish GC-experienced from non-GC-experienced CXCR5⁺CD4⁺ T cells still remains a challenge, it holds the key to addressing the identity of the precursor cells for the circulating CXCR5⁺CD4⁺ T cell population in human peripheral blood.

In summary, our study has identified an important cell surface marker that allows us to distinguish the PD-1⁺CXCR5⁺CD4⁺ T cells that will differentiate into GC-Tfh cells versus CXCR5⁺CD4⁺ memory T cells. CXCR5⁺CD4⁺ memory T cells have been shown to be multi-potent[12]. Thus, the fact that PD-1⁺CXCR5⁺CD4⁺ T cells differentiate into divergent fates with both CXCR5⁻ and CXCR5⁺ CD4⁺ T memory cells having the potential to carry a central memory phenotype will have wide-reaching implications for CD4⁺ T cell responses and diversified CD4⁺ T cell memory. We have also identified that c-Maf, as well as its downstream factor Plekho1, play a stage-specific role in the regulation of GC-Tfh cell differentiation. Understanding the various stages, subsets, and the underlying regulatory mechanisms of Tfh cell differentiation will provide knowledge to help design new strategies for treatment of infectious diseases, autoimmune disorders, and to aid vaccine development for new pandemic threats.

## Methods

### Reagents and resources

See Supplementary Tables 1 to 4 for all reagents and resources.

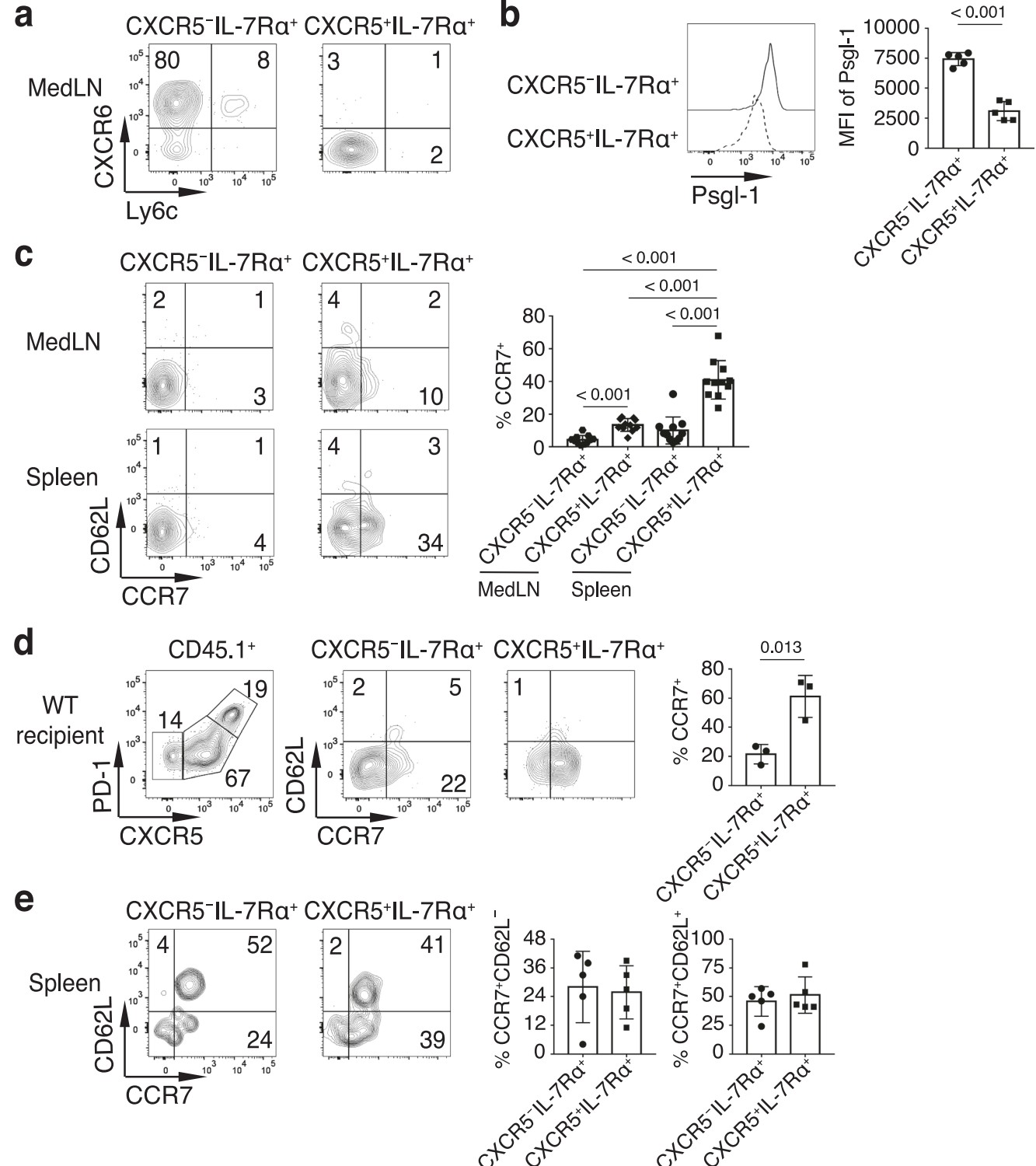

**Fig. 5 | Central memory phenotype induction in CXCR5⁻ and CXCR5⁺ IL-7Rα⁺CD4⁺ T cells.** a–c Purified OT-II cells were transferred into CD45.1⁺ SMARTA recipient mice followed by intranasal infection with PR8-OVA. Indicated donor cell populations at day 28 p.i. in the medLN were analyzed for CXCR6 and Ly6c staining (**a**) and Psgl-1 staining (**b**). **c** Indicated donor OT-II populations at day 28 p.i. in the medLN and spleen were analyzed for CD62L and CCR7 staining (n = 11). **d** Purified CD45.1⁺ OT-II cells were transferred into CD45.2⁺ C57BL/6 recipient mice followed by intranasal infection with PR8-OVA. Indicated donor OT-II populations at day 14

p.i. in the medLN were analyzed for CD62L and CCR7 staining (n = 3). **e** C57BL/6 mice were intranasally immunized with KLH-GP$_{61\text{-}80}$ and LPS. Indicated GP⁺ populations at day 35 p.i. in the spleen were analyzed for CD62L and CCR7 staining (n = 5). Data in **a**–**e** are representative (or pooled) results of at least two independent experiments. Bars represent average ±SD. The *P*-values were determined by a two-tailed unpaired *t*-test (**b, d, e**), or a one-way ANOVA with Tukey's multiple comparisons test (**c**). Source data are provided as a Source Data file.

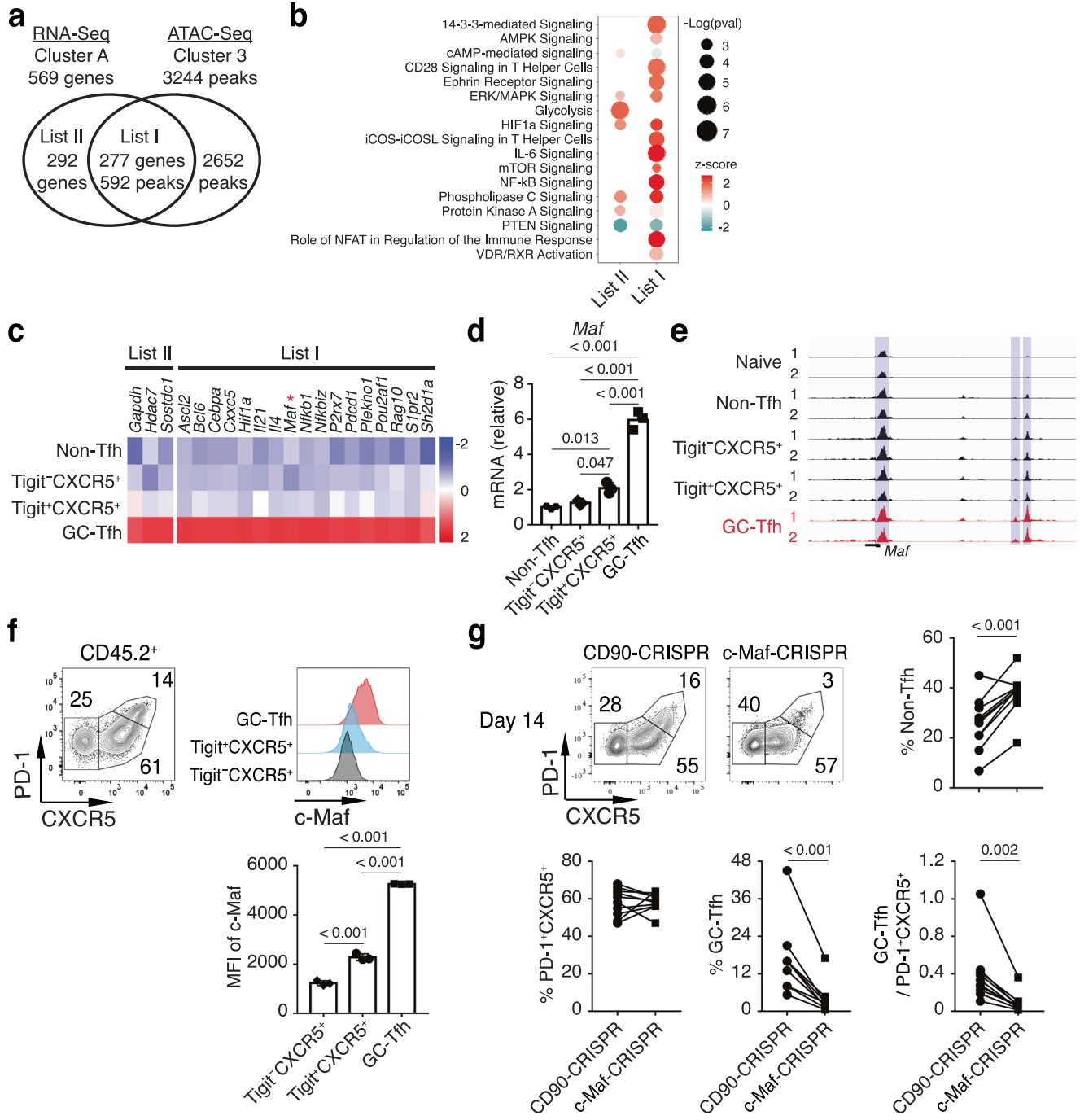

**Fig. 6 | c-Maf is a critical regulator of pre- to GC-Tfh cell differentiation. a** Venn diagram of genes in RNA-seq Cluster A and gene loci in ATAC-seq Cluster 3 from Figs. 3a and 3d, respectively. **b** IPA of List I and List II genes from Fig. 6a. **c** Expression heatmap of select genes from List I and List II. **d** Donor OT-II cells at day 14 p.i. in the medLN of PR8-OVA-infected recipient mice were analyzed by real time RT-PCR for *Maf* mRNA levels (*n* = 3). **e** ATAC-seq read density at the *Maf* gene locus of naïve and indicated donor cell populations as in Fig. 3d. **f** Purified OT-II cells were transferred into CD45.1⁺ SMARTA recipient mice followed by intranasal infection with PR8-OVA. Donor OT-II cells at day 14 p.i. in the medLN were analyzed for PD-1 and CXCR5 staining (left), and indicated populations were analyzed for

c-Maf intracellular staining (right, *n* = 3). **g** CRISPR/Cas9 was used to delete *Thy1* (CD90-CRISPR) or *Maf* (c-Maf-CRISPR) in purified OT-II cells from CD45.1⁺CD45.2⁺ OT-II^Tg or CD45.2⁺ OT-II^Tg mice, respectively. CD90-CRISPR and c-Maf-CRISPR OT-II cells were co-transferred into CD45.1⁺ SMARTA recipient mice followed by intranasal infection with PR8-OVA. Donor OT-II cells at day 14 p.i. in the medLN were analyzed for PD-1 and CXCR5 staining (*n* = 10). Data in **d**, **f**, **g** are representative (or pooled) results of two independent experiments. Bars represent average ±SD. The *P*-values were determined by a two-tailed paired *t*-test (**g**), or a one-way ANOVA with Tukey's multiple comparisons test (**d**, **f**). RNA-seq and ATAC-seq samples are independent biological replicates. Source data are provided as a Source Data file.

## Mice

CD45.2⁺ C57BL/6 (Strain #: 000664), CD45.1⁺ C57BL/6 (Strain #: 002014), OT-II (Strain #: 004194), CD45.1⁺ SMARTA (Strain #: 030450), B1-8i (Strain #: 012642), and CD4-Cre^Tg (Strain #: 017336) mice were purchased from Jackson Laboratories. CD45.2⁺ C57BL/6 mice were bred to CD45.1⁺ C57BL/6 congenic mice to generate CD45.1⁺CD45.2⁺ mice. OT-II mice were bred to CD45.1⁺ C57BL/6 congenic mice to generate CD45.1⁺CD45.2⁺OT-II^Tg and CD45.1⁺OT-II^Tg mice. Fucci2-cell cycle-reporter mice (Strain #: RBRC06511) were obtained from RIKEN and bred with OT-II mice to generate

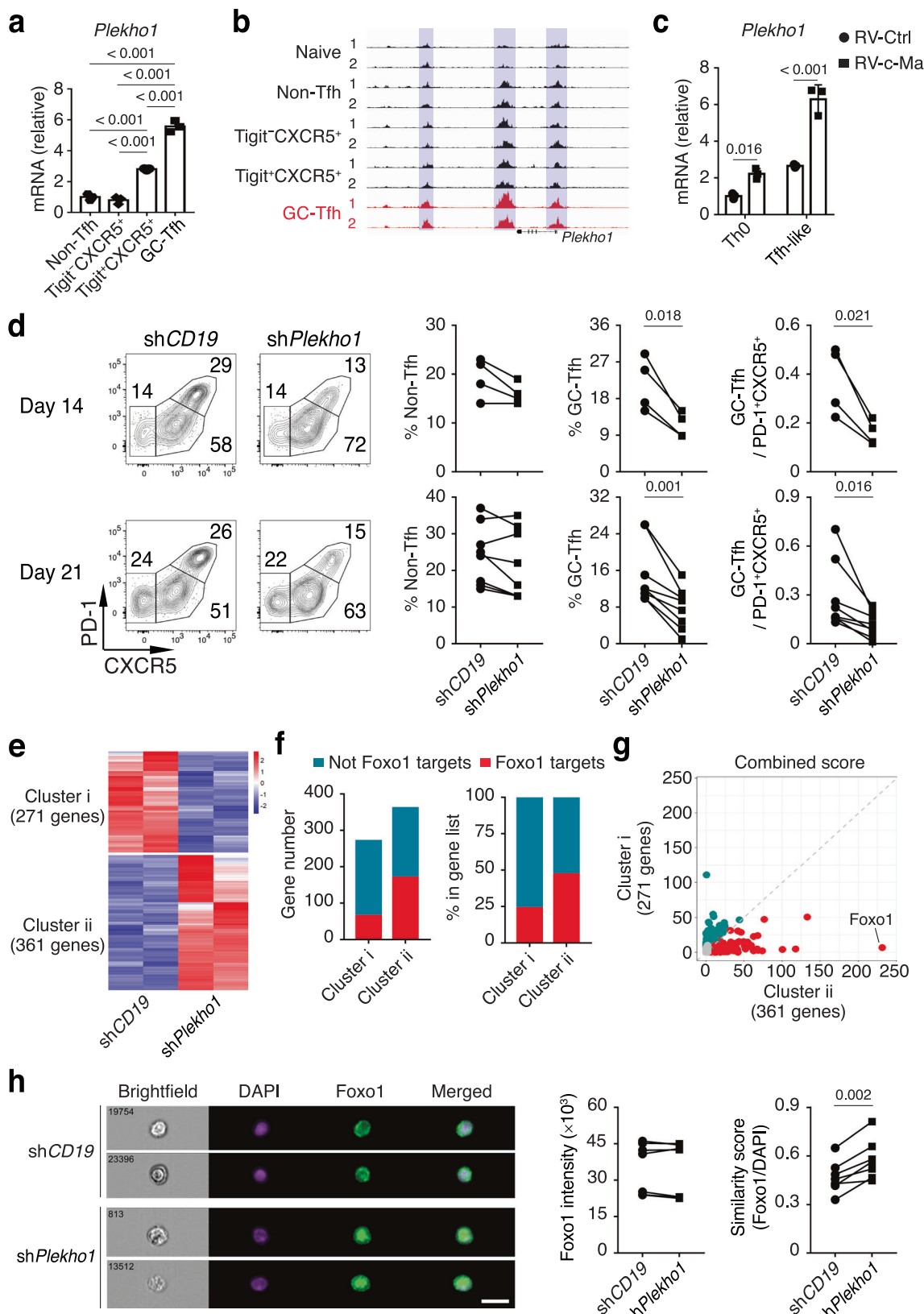

OT-II[Tg]Fucci2 mice. Bcl6[f/f] mice were generated at TSRI Mouse Genetics Core[62]. Bcl6[f/f] mice were bred with CD4-Cre[Tg] and OT-II mice to generate OT-II[Tg]Bcl6[f/f]CD4-Cre[Tg] mice. Bcl6[f/f] mice were bred with CD4-Cre[Tg] and CD45.1[+] C57BL/6 congenic mice to generate CD45.1[+]Bcl6[f/f]CD4-Cre[Tg] mice. Bcl6-protein reporter mice were generated at the Faculty of Pharmaceutical Sciences, Tokyo University of

Science[31]. Experiments were conducted using age and sex matched male and female mice at 6-12 weeks of age. All animals were maintained in specific pathogen–free barrier facilities and were used in accordance with protocols approved by the Institutional Animal Care and Use Committee of the University of Alabama at Birmingham (Birmingham, Alabama).

**Fig. 7 | Plekho1 plays a stage-specific role in regulating the competitive fitness of GC-Tfh cells. a** Purified OT-II cells were transferred into CD45.1[+] SMARTA recipient mice followed by intranasal infection with PR8-OVA. Indicated donor cell populations at day 14 p.i. in the medLN were analyzed by real time RT-PCR for *Plekho1* mRNA levels ($n = 3$). **b** ATAC-seq read density at the *Plekho1* gene locus of naïve and indicated donor cell populations as in Fig. 3d. **c** Purified CD4[+] T cells from C57BL/6 mice were activated in vitro and infected with control retrovirus (RV-Ctrl) or retrovirus expressing *Maf* (RV-c-Maf) under Th0 and Tfh-like culture conditions. On day 4, retrovirally infected cells were sorted and analyzed by real time RT-PCR for *Plekho1* mRNA levels ($n = 3$). **d–h** Purified OT-II cells from CD45.1[+]CD45.2[+] and CD45.2[+] OT-II[Tg] mice were activated in vitro and infected with retrovirus expressing shRNAmir against *CD19* (sh*CD19*, CD45.1[+]CD45.2[+]) and *Plekho1* (sh*Plekho1*, CD45.2[+]). Retrovirally infected OT-II cells were co-transferred into CD45.1[+] SMARTA recipient mice followed by intranasal infection with PR8-OVA two days after cell

transfer. **d** Donor OT-II cells at days 14 and 21 p.i. in the medLN were analyzed for PD-1 and CXCR5 staining (day 14, $n = 4$; day 21, $n = 8$). **e–g** Donor OT-II cells at day 14 p.i. in the medLN were analyzed by RNA-seq. **e** Clustered heatmap of sh*CD19* and sh*Plekho1* GC-Tfh cells. **f** Numbers and percentages of potential Foxo1 targets in Cluster i and Cluster ii genes from Fig. 7e. **g** Enrichr enrichment analysis of transcription factors associated with DEGs in Fig. 7e. **h** ImageStream analysis of Foxo1 sublocation in GC-Tfh cells 30 min after stimulation with anti-CD3/ICOS antibodies ($n = 7$). Scale bars, 12 μm. Data in **a**, **c**, **d**, **h** are representative (or pooled) results of two independent experiments. Bars represent average ±SD. The *P*-values were determined by a two-tailed paired *t*-test (**d**, **h**), a one-way ANOVA with Tukey's multiple comparisons test (**a**), or a two-way ANOVA with Sidak's multiple comparisons test (**c**). RNA-seq and ATAC-seq samples are independent biological replicates. Source data are provided as a Source Data file.

## Adoptive transfer, infection, and immunization

OT-II cells were purified from the lymph nodes and spleens using the Invitrogen Dynabeads CD4 positive isolation kit (Thermo Fisher Scientific) or mouse CD4 (L3T4) MicroBeads (Miltenyi Biotec; >95% CD44[lo]CD62L[hi]). For OT-II cell transfer experiments, $3 \times 10^4$ purified OT-II cells were transferred intravenously into SMARTA or C57BL/6 recipient mice followed by influenza virus infection. For sh*Tigit* experiments, B cells were purified from the lymph nodes and spleens using mouse CD43 (Ly-48) MicroBeads (Miltenyi Biotec). In all, $5 \times 10^5$ purified B1-8i cells with $3 \times 10^5$ sorted Ametrine[+] OT-II cells were co-transferred intravenously into CD45.1[+]*Bcl6*[f/f]CD4-Cre[Tg] recipient mice, rested for 1 day, followed by intranasal immunization with 100 μg NP-OVA and 3 μg LPS. For sh*Plekho1* experiments, $2 \times 10^5$ retrovirally infected OT-II cells were co-transferred with sh*CD19* control OT-II cells intravenously into CD45.1[+] SMARTA recipient mice followed by intranasal infection with PR8-OVA 2 days after cell transfer. For Tigit[+]IL-7Rα[−] and Tigit[−]IL-7Rα[+] co-transfer experiments, $2 \times 10^4$ Tigit[+]IL-7Rα[−] and $2 \times 10^4$ Tigit[−]IL-7Rα[+] cells were sorted at day 14 p.i. from the mediastinal lymph nodes of CD45.1[+]CD45.2[+] or CD45.2[+] C57BL/6 mice infected with PR8, mixed at 1:1 ratio, and co-transferred intravenously into CD45.1[+] C57BL/6 recipient mice followed by intranasal infection with PR8 at the same time of transfer. For c-Maf-CRISPR or Plekho1-CRISPR co-transfer experiments, $3 \times 10^4$ c-Maf-CRISPR or Plekho1-CRISPR OT-II cells were co-transferred with $3 \times 10^4$ CD90-CRISPR OT-II cells intravenously into SMARTA recipient mice followed by influenza virus infection. For Plekho1-CRISPR and CD90-CRISPR separate transfer experiments, $5 \times 10^5$ Plekho1-CRISPR OT-II cells and $5 \times 10^5$ CD90-CRISPR OT-II cells were transferred intravenously into CD45.1[+]*Bcl6*[f/f]CD4-Cre[Tg] recipient mice, followed by influenza virus infection. For all influenza virus infection, mice were immobilized with isoflurane and intranasally infected with mouse-adapted influenza virus A/Puerto Rico/8/34 (PR8) at a dose of 15,000 virus forming units (v.f.u.) or PR8 virus expressing the OVA (323-339) epitope (PR8-OVA) at a dose of 200–1000 v.f.u. For GP[61]-KLH/LPS immunization, mice were immobilized with isoflurane and intranasally immunized with 20 μg GP[61]-KLH and 3 μg LPS.

## Cell preparation, staining, flow cytometry, and ImageStream

Mediastinal lymph nodes and spleens were mashed through a 70 μm filter to obtain single cell suspensions. Spleen samples were further incubated with ACK lysis buffer to remove red blood cells. Cells were counted using trypan blue staining, and $3 \times 10^6$ to $5 \times 10^6$ cells were suspended in 50 μl phosphate buffered saline (PBS) containing 2% bovine serum albumin and 2 mM EDTA (FACS buffer) for staining. Nonspecific antibody binding was blocked with anti-CD16/CD32 antibodies (Biolegend) in FACS buffer for 10 min before staining. Dead cells were excluded through the use of a Live/Dead Fixable Dead Cell staining kit or Fixable Viability Dye eFluor 780 (Invitrogen). Allophycocyanin (APC) conjugated influenza nucleocapsid protein (NP[311-325]) tetramer and lymphocytic choriomeningitis virus glycoprotein (GP[66-77]) tetramer were provided by the NIH Tetramer Core Facility, and the staining were

performed on room temperature (RT) for 1 h. For intracellular Bcl6 and c-Maf staining, cells were fixed and permeabilized using the eBioscience Foxp3 transcription factor staining kit (Thermo Fisher Scientific). Intracellular staining was performed on ice for 45 min. For intracellular Foxo1 staining, cells were stimulated with 5 μg/ml anti-ICOS (Biolegend) and 0.5 μg/ml anti-CD3 (eBioscience) antibodies followed by cross-linking with 20 μg/ml goat anti-hamster IgG (MP Biomedicals) at 37 °C for 30 min. After surface staining, cells were fixed and permeabilized using the BD Cytofix/Cytoperm™ Fixation/Permeabilization solution kit (BD Biosciences). The primary anti-Foxo1 antibody (Cell Signaling Technology) staining was performed at RT for 1 h, and it was followed by secondary anti-rabbit IgG (minimal x-reactivity) DyLight 488 antibody (Biolegend) staining on ice for 45 min. Samples were analyzed or sorted using custom BD LSR II, BD FACSymphony, BD FACSAria II, or Amnis ImageStream[X] Mark II instruments in the UAB Comprehensive Flow Cytometry Core Facilities. Flow cytometry results were analyzed using FlowJo software (v.10.7.1). Imaging flow cytometry results of 10,000–30,000 cells each mouse of two independent experiments (total 7 mice) were analyzed using IDEAS software (v.6.2), and the nuclear intensity of Foxo1 reflects the amount of Foxo1 within the DAPI nuclear mask. For quantification, the fluorescence intensities of all acquired cells of each mouse were averaged.

## CRISPR/Cas9-mediated knockout in OT-II cells

Three unique guides specific to *Maf* and *Plekho1* were obtained according to IDT's online predesigned library (https://www.idtdna.com/site/order/designtool/index/CRISPR_PREDESIGN) specifically targeting the 5′ end of *Maf* and *Plekho1*. For further details, see ref. 63.

## Immunofluorescence staining and microscopy

Mediastinal lymph nodes were harvested, pre-fixed with 4% paraformaldehyde at RT for 30 min, followed by 10% sucrose at RT for 2 h, and 30% sucrose at 4 °C for overnight. Dehydrated samples were embedded in optimal cutting temperature (OCT) medium (Tissue-Tek) for cryosectioning. Cryosectioning was performed on a Leica CM1800 cryostat. Tissue sections (20 mm-thick) were mounted on glass slides and stored at −20 °C. For staining, frozen sections were thawed at RT for 30 min, fixed in acetone at RT for 10 min, and air dried at RT for 10 min. Sections were rehydrated by soaking in PBS at RT for 5 min, blocked with PBS containing 10% normal goat serum (Abcam) at RT for 20 min, then incubated with antibodies in staining buffer (Earle's balanced salt solution [EBSS] with 1% Bovine serum albumin and 0.1% Triton X-100, filtered before use) in the dark. Anti-rabbit IgG (minimal x-reactivity) DyLight 488 (Biolegend), anti-mouse CD4 BV421 (Biolegend), anti-mouse CD45.2 PE (eBioscience), anti-mouse/human GL7 PB (Biolegend), anti-mouse/human GL7 AF488 (Biolegend), anti-mouse IgD AF647 (Biolegend) and anti-mouse Tigit (R&D) were used. To prevent autofluorescence, stained sections were incubated with 0.05% Sudan Black B (dissolved in ethanol, freshly prepared and filtered before using) at RT for 10 min in the dark, then washed with PBS for

5 min at RT three times. Stained sections were mounted with Pro-Long™ Gold Antifade Mountant without DAPI (Thermo Fisher Scientific) and sealed with a glass coverslip. Confocal microscopy images were acquired using a Nikon A1R confocal microscope, and representative sections were acquired at ×4 to ×20 magnification.

## T cell stimulation and retroviral transduction

Purified CD4⁺ T cells were stimulated for 48 h with 0.5 µg/ml anti-CD3 (eBioscience) and 1 µg/ml anti-CD28 (eBioscience) antibodies in plates pre-coated with 20 µg/ml goat anti-hamster IgG (MP Biomedicals) in complete T cell medium (Dulbecco's Modified Eagle's Medium [DMEM], supplemented with 10% heat-inactivated FBS, 2 mM L-glutamine, penicillin-streptomycin, nonessential amino acids, sodium pyruvate, vitamins, 10 mM HEPES, and 50 µM 2-mercaptoethanol), then their populations were expanded for another 2 days in complete T cell medium containing 20 U/ml recombinant human IL-2. For the sh*Tigit* and sh*Plekho1* knockdown experiments, retrovirus was produced by Plat-E cells co-transfected with retroviral vectors and helper plasmids via PEI (Polysciences) mediated transduction. Activated OT-II cells were transduced at 18 and 48 h with virus-containing media supplemented with 0.6 µg/ml polybrene (Sigma). After the second transduction, cells were expanded for 2 days in complete T cell medium containing 20 U/ml recombinant human IL-2. For the c-Maf overexpression experiments, retrovirus was produced by Plat-E cells co-transfected with retroviral vectors and helper plasmids via PEI (Polysciences) mediated transduction. Activated OT-II cells were cultured under Th0 or Tfh-like condition and transduced at 18 and 48 h with virus-containing media supplemented with 0.6 µg/ml polybrene (Sigma). After the second transduction, cells were expanded with same condition for 2 days. Th0 condition: 20 U/ml rhIL-2; Tfh-like condition: 10 ng/ml IL-6, 10 ng/ml IL-21, 10 µg/ml anti-IL-4, 10 µg/ml anti-IFN-γ, and 10 µg/ml anti-TGF-β.

## Real-time RT-PCR

RNA from sorted donor cell populations was purified, and real-time reverse-transcription was performed as described[64]. Expression of mRNA was normalized to *Rpl32* expression.

## RNA sequencing

Cells were sorted directly into TRIzol-LS Reagent (Invitrogen) and RNA was isolated using miRNeasy Micro Kit (QIAGEN). Illumina library preparation and sequencing were performed at Scripps Research Institute Next Generation Sequencing Core facility (La Jolla, California) or GENEWIZ. Data analysis was performed using the Cheaha Supercomputer at the University of Alabama at Birmingham. Low quality reads and sequencing adapters were removed using Trim Galore! (v.0.4.4). Trimmed reads were aligned to the mouse genome assembly mm10 using STAR (v.2.5.4b) with genes counts determined by using HTSeq-Count (v.0.12.3) or aligned to the mouse genome assembly mm10 and quantified using Salmon (v.0.14.1). Further analyses were performed with R software (v.4.0.3). Raw read counts were normalized, and differential expression was analyzed using the Bioconductor package DESeq2 (v.1.28.1). PCA was performed on the top 500 variable genes (normalized with rlog function) using the plotPCA function in DESeq2. Heatmaps were generated with the package pHeatmap (v.1.0.12) using genes with an adjusted *p* value <0.01, and clusters were generated using Ward's method. Venn diagram was built using genes with an adjusted p value less than 0.05. IPA (v.60467501; Ingenuity Systems; Qiagen, Redwood City, California) was performed using DEGs between Tigit⁺PD-1⁺CXCR5⁺ and Tigit⁻PD-1⁺CXCR5⁺ cells, Cluster A and B genes comparing PD-1ʰⁱCXCR5ʰⁱ GC-Tfh and Tigit⁺PD-1⁺CXCR5⁺ cells, or List I and II genes comparing PD-1ʰⁱCXCR5ʰⁱ GC-Tfh and Tigit⁺PD-1⁺CXCR5⁺ cells. Pathway enrichment analysis was performed using DEGs between sh*Plekho1* and sh*CD19* GC-Tfh cells on the website of Metascape (https://metascape.org/gp/index.html#/main/

step1, default parameters). GSEA (v.4.1.0) was performed using the Broad Institute software (https://www.broadinstitute.org/gsea/index.jsp) and the enrichment scores (default parameters) were calculated by comparing the indicated groups. Enrichment analysis of transcription factor association was performed using the ChEA2022 module within Enrichr (https://maayanlab.cloud/Enrichr/, default parameters).

## ATAC sequencing

In all, $5 \times 10^4$ cells were sorted into 1.5 ml tubes and ATAC-seq library was prepared as described[65]. Sequencing was performed at Scripps Research Institute Next Generation Sequencing Core facility (La Jolla, California). Data analysis was performed using the Cheaha Supercomputer at the University of Alabama at Birmingham. Low quality reads and sequencing adapters were removed using Trim Galore! (v.0.4.4). Trimmed reads were aligned to the mouse genome mm10 using Bowtie2 (v.2.3.3). PCR duplicates were removed using Picard (v.2.20.0). Peak calling was performed using MACS v2 (FDR q-value 0.01). IDR (v.2.0.3) software was used to find reproducible peaks between sample replicates. For each experiment, we combined peaks of all samples to create a union peak list and merged overlapping peaks with BedTools (v.2.28.0) merge. Further analyses were performed with R software (v.4.0.3). DARs were identified following DESeq2 (v.1.28.1) normalization using an adjusted p value less than 0.05. PCA was performed on the top 500 most variable peaks (normalized by rlog function) using the plotPCA function in DESeq2. The heatmap was generated using the package pHeatmap (v.1.0.12) using peaks with an adjusted *p* value <0.01, and clusters were generated using Ward's method. Motif enrichment was calculated using HOMER (v.4.11.1, default parameters) on peaks within indicated clusters. The read density of gene loci was visualized and generated using IGV (v.2.7.2).

## Statistics

Independent and paired two-tailed Student's *t* tests, independent and repeated measure one-way ANOVAs, and repeated measure two-way ANOVAs were performed using GraphPad Prism software (v.8.2.1). All error bars represent standard deviation. Though appropriate statistical tests were run for all comparisons made, *P*-values were only displayed for significant results. For further details, see figure legends.

## Reporting summary

Further information on research design is available in the Nature Portfolio Reporting Summary linked to this article.

# Data availability

RNA-seq and ATAC-seq data generated in this study have been deposited at GEO (SuperSeries accession number: GSE174104). The authors declare that the data supporting the findings of this study are available within the paper and its supplementary information files. Source data are provided with this paper.

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

## Acknowledgements
We thank Marion Spell, Vidya Sagar Hanumanthu, Harish Chandra Pal, Dr. Ana Longhini, Berencia Fore, Clarisse Iradukunda, Karlee Stortz, and Teshia Houser for flow cytometry and cell sorting, Dr. Baiyi Cai, Dr. Shuaiwei Wang (University of Oxford), Shreya Kashyap, Kim Nguyen and Ching-En Lee for assistance with histology, Drs. Frances Lund and Troy Randall (University of Alabama at Birmingham) for the PR8 and PR8-OVA influenza viruses, and Drs. Remy Bosselut and Avinash Bhandoola (National Institutes of Health/NCI) for discussion. This work was supported by National Institutes of Health Grants AI130232 and AI167986 (to H.H.), AI122842, AI116188, and AI133679 (to O.K.), NIH training grants T32AR069516 (to R.J.M.), T32AI007051 (to A.R.S.), and T32GM008361 (to R.J.M. and A.R.S.), and University of Alabama at Birmingham Center for AIDS Research Grant P30AI027767-26.

## Author contributions
R.J.M., F.Z., A.R.S., X.X., and Y-H.W. designed and performed experiments; D.O.S., K.M.C. D.F., and E.R.B. helped with experiments; A.J.G. generated shRNAmir targeting under the supervision of M.E.P.; F.Z., B.D.G., E.G.-A., and D.O.S. performed bioinformatics analyses under the supervision of H.H. and O.K.; C.X. and Y.H. provided key reagents; H.H. conceived the study and provided overall supervision; and F.Z., R.J.M., A.R.S., and H.H. wrote the paper.

## Competing interests
The authors declare no competing interests.
