## [Peer Review File · Nature Communications]

Spatiotemporal resolution of germinal center Tfh cell differentiation and divergence from central memory CD4⁺ T cell fateREVIEWER COMMENTS

Reviewer #1 (Remarks to the Author):

Summary

This study addresses important questions regarding the differentiation of T follicular helper cells (Tfh) using mouse models of influenza infection and analyses of disaggregated mediastinal lymph node and spleen cells. In the first portion of the manuscript, persistent TIGIT expression is identified as a marker of cells destined to become germinal center (GC) TFH. TIGIT is shown not to be required for PD-1^{hi}CXCR5^{hi} GC TFH differentiation, but it is expressed (and re-expressed) in the context of TCR stimulation suggesting that B cells presenting antigen to TFH either at the T-B border of follicles or in GC may be important in this differentiation process. Using transcriptomic analyses, distinct gene expression profiles were demonstrated for non-TFH, TIGIT-CXCR5+PD-1+, TIGIT+CXCR5+PD-1+, and GC TFH; TIGIT+CXCR5+PD-1+ cells appeared to be closely related to TIGIT-CXCR5+ PD-1+, and most closely related to GC TFH. Chromatin access analyses revealed significant differences between TIGIT+CXCR5+PD-1+ cells and GC TFH, suggesting that additional differentiation events must occur for TIGIT+CXCR5+PD-1+ cells to differentiate into GC TFH. IL-7Ra was not expressed on TIGIT+ cells and vice versa. Furthermore, adoptive transfer of these subpopulations into mice revealed that TIGIT-CXCR5+PD-1+ cells did not evolve into GC TFH, but instead became memory cells, whereas the TIGIT+CXCR5+PD-1+ cells resulted in a robust GC TFH response. From these data the authors conclude that TIGIT-CXCR5+PD-1+ cells become memory CD4+ T cells, whereas TIGIT positive cells are the precursor to GC TFH. Furthermore, they conclude that the competitiveness of the environment as well as the tissue (nondraining lymphoid organ, e.g., spleen, vs draining lymphoid organ, e.g., lymph node) are important determinants of the pathway that the pre-GC TFH will take. The authors next examined the role of c-Maf in differentiation of GC TFH. They selected c-Maf because it has previously been shown to be important for TFH differentiation, but the underlying mechanism is not understood. Furthermore, in their assays of PD-1+CXCR5+CD4+ T cells and GC TFH, c-Maf had increased mRNA expression as well as chromatin accessibility. Additional studies confirmed that c-Maf deletion by CRISPR/Cas9 did not impair CXCR5+CD4+ T cell generation, but did block GC TFH development, suggesting that c-Maf is critical in the transition from pre- to GC-TFH transition. Over-expression of c-Maf increased expression of Plekho1 and Cxhc5, and inhibition of these factors by shRNAmir revealed that non-TFH evolved well, but there was an impairment in differentiation of PD-1+CXCR5+ TFH cells to GC TFH.

Review

The use of mouse models of influenza infection in this paper is elegant. The manuscript is well written, the tables and figures are appropriate and well done. Identification of TIGIT as a marker for GC TFH precursors could be quite useful to the field. Furthermore, establishing that TIGIT-TFH become memory cells is also a contribution. Understanding the factors that promote differentiation of GC TFH is of great potential importance, given their major role in many infectious and autoimmune diseases. The findings of this study will naturally lead to additional studies to parse out the roles of Plekho1 and Cxhc5 with the potential to manipulate them to address various diseases or promote vaccine strategies. Overall, the findings are convincing and they represent important contributions to the field. A few suggestions:

- 1) It would substantially strengthen the manuscript if in situ analyses of lymph node and spleen tissues from these animals were provided, demonstrating the physical location of these various cell subsets, i.e., TIGIT+ and TIGIT- CXCR5+PD-1+ CD4+ cells, as well as TIGIT+ GC TFH and IL-7Ra+ subsets. Are the TIGIT+CXCR5+PD-1+ cells mainly located at the T-B border, or are they in the follicles or GC? Does the location of TIGIT+ cells change over time? In the c-Maf experiments, it would be interesting to demonstrate the localization of c-Maf+ cells, and furthermore to show that GC development is blocked by inhibitors of Plekho1 and Cxhc5.
- 2) A limitation of using disaggregated cells alone is that they only provide percentages, but not absolute numbers or counts. Few experiments reported used absolute cell counts. This limitation can be overcome by the tissue analyses suggested above where the size of the lymph node or spleen can be measured (or determined by weight) and some idea of the actual magnitude of GC TFH development determined.
- 3) The manuscript seems a little disjointed in that TIGIT is the focus of the first half and c-Maf is the focus of the second half, with little unifying these two sections. It would be interesting to know

how over-expression of c-Maf affected TIGIT and IL-7Ra expression. Similarly, it would further strengthen the value of these markers if it was shown that inhibition of Plekho1 and Cxcr5 also reduced expression of TIGIT and increased IL-7Ra.

4) The true nature of circulating TFH is a matter of great controversy. If PBMC are available, it would be interesting to know whether cTFH express TIGIT, IL-7Ra, and c-Maf.

Reviewer #2 (Remarks to the Author):

Zhu et al. in this study analyzed the phenotypical and functional heterogeneity of PD-1+CXCR5+CD4+ T cells developed in lymphoid organs after viral infection or immunizations. The study shows that TIGIT+ compartment within PD-1+CXCR5+CD4+ T cells displayed a tendency to mature into GC-Tfh cells, while TIGIT- counterpart tended to become central memory cells upregulating the expression of IL-7R +/- CCR7. The analysis of DEGs and DARs between PD-1+CXCR5+CD4+ T cells and GC-Tfh cells has led to the discovery that cMaf plays an important role in GC-Tfh cell differentiation. Furthermore, expression of Plekho1 and Cxcr5 was upregulated by cMaf in in-vitro generated Tfh-like cells, and both factors were required for optimal GC-Tfh cell differentiation in vivo.

Overall, the experiments were well performed, and the data are convincing. This study has shown the subsets within CXCR5+PD-1+ cells with different cell differentiation fates – TIGIT+IL-7R- cells and TIGIT-IL-7R+ cells. Although TIGIT expression did not affect GC-Tfh cell differentiation per se, this study shows that TIGIT and IL-7R are both useful markers to define the precursors of GC-Tfh cells and CM cells.

I have only a few comments.

In Fig. 4B: The flow panel of CXCR5 vs. TIGIT for PD-1+CXCR5+ population will be informative to show the relationship between TIGIT and CXCR5 expression levels, and the time course of TIGIT expression levels.

Weak T cell response (L210): This term is very vague, and I am not sure what the term T cell response implicates here. TCR signal strength? Environmental factors such as cytokines? Antigen amounts?

Reviewer #3 (Remarks to the Author):

In the paper "The divergence of PD-1+CXCR5+CD4+ T cell differentiation and stage-specific regulation of GC-Tfh cells", Zhu F et al. applied several mouse models to characterize Tigit expression and function between Tfh (PD-1+CXCR5+) and GC-Tfh (PD-1hiCXCR5hi) cells. Authors demonstrate that, although Tigit expression is sustained in GC-Tfh cells, it is not required for Tfh or GC-Tfh generation or function. In addition, authors report c-Maf expression increases along Tfh to GC-Tfh differentiation and is required for the optimal GC-Tfh differentiation, with the underlying mechanism for the latter suggested, partially through the induction of Plekho1 and Cxcr5. I appreciate the large amount of data presented by the authors. Although the analyses are clear and largely convincing, the study is not coherent and doesn't deliver significant advancement due to several conceptual issues.

First of all, the first part of the study on Tigit (Fig1-4) is not connected with either memory phenotype (Fig5) or c-Maf/Plekho1 and Cxcr5 (Fig6-7). Each of the parts is not fully completed, as I will describe below.

Next, there are two major questions in the Tigit part. First, Tigit was upregulated in all antigen-specific cells at day 7 (early phase) in several models and then was downregulated on some cells at late time points. The central message claimed by the authors -- "that the sustained Tigit expression in PD-1+CXCR5+CD4+ T cells marked the precursor Tfh (pre-Tfh) to GC-Tfh transition" (abstract). This is not entirely supported by the data. Although the authors didn't show, there were mixed non-Tfh, Tfh and GC-Tfh at day 7 in these models. At least at day 7, Tigit expression didn't distinguish Tfh vs GC-Tfh and didn't mark pre-Tfh to GC-Tfh transition. There is at least another possibility for the different Tigit expression between Tfh vs GC-Tfh at late time points. GC-Tfh

might exit GC and become memory cells with the downregulation of Tigit. Several previous studies indicate that pre-Tfh to GC-Tfh transition and the divergence of differentiation happen early in the response, usually before day 7 (Kerfoot SM 2011; Choi YS 2011; He J 2013). It is not very suitable to address the question by focusing on 2 or 3 weeks after immunization or infection. Second, authors compared Tigit+ vs Tigit- CXCR5+ population. As Tigit expression and CXCR5 expression are positively correlated (Fig 1F), I assume Tigit- CXCR5+ population expresses high CXCR5 than Tigit- CXCR5+ population. Therefore, the former is a partial GC-Tfh cells. The RNA-seq and ATAC-seq results were largely expected.

Last, c-Maf's function in both mouse and human Tfh generation and function have been extensively studied and reported. The two candidate targets of c-Maf-mediated regulation, Plekho1 and Cxcr5 are potentially interesting, but the characterization was superficial. Besides changing GC-Tfh percentages, it would be necessary to know how they might regulate T cell proliferation, survival or other T effector differentiation, as well as any Tfh functional alternation.

Overall, all parts of the paper are unable to draw exciting conclusions with significant advancement in the field.

NCOMMS-22-39526: The divergence of PD-1⁺CXCR5⁺CD4⁺ T cell differentiation and stage-specific regulation of GC-Tfh cells

Reviewer #1 (Remarks to the Author):

1. It would substantially strengthen the manuscript if *in situ* analyses of lymph node and spleen tissues from these animals were provided, demonstrating the physical location of these various cell subsets, i.e., TIGIT⁺ and TIGIT⁻ CXCR5⁺PD-1⁺ CD4⁺ cells, as well as TIGIT⁺ GC TFH and IL-7Ra⁺ subsets. Are the TIGIT⁺CXCR5⁺PD-1⁺ cells mainly located at the T-B border, or are they in the follicles or GC? Does the location of TIGIT⁺ cells change over time? In the c-Maf experiments, it would be interesting to demonstrate the localization of c-Maf⁺ cells, and furthermore to show that GC development is blocked by inhibitors of Plekho1 and Cxyc5.

Response: We thank the Reviewer for these important questions. We have been testing almost all the commercially available anti-Tigit antibodies, and we recently identified one suitable for histology staining. Combining histology and confocal microscopy, we first demonstrated that in the influenza virus infection model, although almost all NP⁺CD44^{hi}CD4⁺ T cells expressed Tigit by day 6-7 p.i. (Supplementary Fig. 1c), there was no GC formation yet (**new Fig. 1b**). The Tigit⁺ donor cells were still mainly in the T cell zone and started to accumulate at the T-B border (**new Fig. 2a**). By day 14 p.i., whereas most of the donor cells in the GCs were Tigit⁺, it appeared that the donor cells inside the B cell follicles were also Tigit⁺ (**new Fig. 2b**); some Tigit⁺ donor cells were found along the T-B border and in the T cell zone (**new Fig. 2b**). The new data are consistent with the results of the kinetics of Tigit expression pattern in non-Tfh, PD-1⁺CXCR5⁺CD4⁺ T cells and GC-Tfh cells, and support that, from the angle of cell localization, the sustained Tigit expression seems to mark the PD-1⁺CXCR5⁺CD4⁺ T cells that differentiate further into the GC-Tfh direction than their Tigit⁻ counterparts.

Although our anti-IL-7R α antibody and anti-c-Maf antibody worked well for cell surface staining and intracellular staining, respectively (Figs. 4 and 6f), unfortunately, they were not suitable for histology staining. Nevertheless, this won't change the results and conclusions that Tigit⁻PD-1⁺CXCR5⁺CD4⁺ T cells upregulate IL-7R α (Fig. 4 and Supplementary Fig. 5) and pre- to GC-Tfh cell transition is severely blocked in the absence of c-Maf (Fig. 6g). Regarding the GC B cell responses with suppressed Plekho1 expression in GC-Tfh cells, please see our response to question #6 of Reviewer #3.

2. A limitation of using disaggregated cells alone is that they only provide percentages, but not absolute numbers or counts. Few experiments reported used absolute cell counts. This limitation can be overcome by the tissue analyses suggested above where the size of the lymph node or spleen can be measured (or determined by weight) and some idea of the actual magnitude of GC TFH development determined.

Response: We agree with the Reviewer that absolute cell numbers are important. The variation in the influenza virus infection model is sometimes considerable. In the KLH-GP₆₁₋₈₀ immunization model, we provided detailed kinetics of PD-1/CXCR5 staining and cell numbers of GC-Tfh cells (**new Supplementary Fig. 5b**), together with the cell numbers of Tigit⁻PD-1⁺CXCR5⁺IL-7R α ⁺ T cells (Supplementary Fig. 5c).

3. The manuscript seems a little disjointed in that TIGIT is the focus of the first half and c-Maf is the focus of the second half, with little unifying these two sections. It would be interesting to know how over-expression of c-Maf affected TIGIT and IL-7Ra expression. Similarly, it would further strengthen the value of these markers if it was shown that inhibition of Plekho1 and Cxyc5 also reduced expression of TIGIT and increased IL-7Ra.

Response: We thank the Reviewer for the suggestion and performed the experiments proposed by the Reviewer. We found that although retroviral expression of c-Maf *in vitro* helped increase the expression levels of Plekho1 (Fig. 7c), the overexpression of c-Maf *in vivo* did not change much the pre- to GC-Tfh

cell differentiation, nor the Tigit or IL-7R α expression in PD-1⁺CXCR5⁺CD4⁺ T cells (**Rebuttal Letter Fig. 1**). These results suggest that the determination of Tigit⁺ versus Tigit⁻PD-1⁺CXCR5⁺CD4⁺ T cells most likely does not involve c-Maf, and therefore the c-Maf overexpression does not have any impact on the Tigit or IL-7R α expression; furthermore, while c-Maf is necessary for GC-Tfh development, it alone is not sufficient to enforce the GC-Tfh program. The knockdown of Plekho1, a factor downstream of c-Maf, also did not seem to affect the Tigit or IL-7R α expression in PD-1⁺CXCR5⁺CD4⁺ T cells (**Rebuttal Letter Fig. 2**).

4. The true nature of circulating TFH is a matter of great controversy. If PBMC are available, it would be interesting to know whether cTFH express TIGIT, IL-7Ra, and c-Maf.

Response: We thank the Reviewer for this very important point. Our study has clearly raised the question about the identity of the precursor cells for the circulating CXCR5⁺CD4⁺ T cell population in human peripheral blood. To push this question further, in human, we will also ask: do PBMC CXCR5⁺CD4⁺ T cells reflect the present or the past of the germinal center responses?

We have been carrying out a series of experiments to address these questions including using a novel GC-Tfh fate-mapping reporter mouse line we recently generated (see our response to question #3 of Reviewer #3). That being said, we hope the Reviewer would agree with us that these points are perhaps better suited for future studies. Meanwhile, we have performed the experiments requested by the Reviewer. We found that in healthy donor PBMC, the naïve IL-7R α ⁺CD45RA^{hi}CD4⁺ T cells did not express much Tigit; neither did the CXCR5⁻ or CXCR5⁺ effector/memory CD4⁺ T cells (**Rebuttal Letter Fig. 3a, b**). While overall, the expression levels of c-Maf in these CD4⁺ T cells were low, the CXCR5⁺CD4⁺ T cells expressed relatively higher levels (**Rebuttal Letter Fig. 3c**).

Reviewer #2

1. In Fig. 4B: The flow panel of CXCR5 vs. TIGIT for PD-1⁺CXCR5⁺ population will be informative to show the relationship between TIGIT and CXCR5 expression levels, and the time course of TIGIT expression levels.

Response: The information requested by the Reviewer is now included in **new Supplementary Fig. 4b** of the revised manuscript.

2. Weak T cell response (L210): This term is very vague, and I am not sure what the term T cell response implicates here. TCR signal strength? Environmental factors such as cytokines? Antigen amounts?

Response: We thank the Reviewer for this important point. In the influenza virus infection model, consistent with the route of intranasal infection, we found that the magnitude of donor T cell expansion in the draining mediastinal LN (medLN) was far greater than that in the spleen (**new Supplementary Fig. 4a**). As the total cell number in the spleen was much higher than that in the medLN, we used the percentage to indicate the degree of expansion (**new Supplementary Fig. 4a**). Based on these results, we reason that most likely due to the available antigen amounts, compared to the response in medLN, the donor T cell response in the spleen (and a competitive environment in wild-type recipient mice) is relatively weaker.

Reviewer #3

1. First of all, the first part of the study on Tigit (Fig1-4) is not connected with either memory phenotype (Fig5) or c-Maf/Plekho1 and Cxxc5 (Fig6-7). Each of the parts is not fully completed, as I will describe below.

Response: We apologize if we haven't clearly explained the link between the Tigit expression pattern and the CXCR5⁺CD4⁺ T cell memory. We have tried to address the question of which CXCR5⁺CD4⁺ T cells

will enter B cell follicle and become GC-Tfh cells while the others may have a different fate. Our results show that it is mainly the Tigit⁻CXCR5⁺CD4⁺ T cells that will upregulate IL-7R α , suggesting that this Tigit⁻ subset of the PD-1⁺CXCR5⁺CD4⁺ T cells differentiates towards the direction of memory formation (Fig. 4). At present, whether CXCR5⁺CD4⁺IL-7R α ⁺CCR7⁺ T cells should be named “Tcm (T central memory)” or not is a hotly debated topic in the CD4⁺ T cell field. In this regard, we have provided data suggesting a mechanism by which CXCR5⁺CD4⁺IL-7R α ⁺ T cells either express CCR7 or do not (Fig. 5).

Regarding c-Maf, we have demonstrated that c-Maf levels progressively increased from Tigit⁻ to Tigit⁺ PD-1⁺CXCR5⁺CD4⁺ T cells and continued to increase in GC-Tfh cells. We think this expression pattern matches and explains the functional outcome of c-Maf such that in the absence of c-Maf, although many PD-1⁺CXCR5⁺CD4⁺ T cells are still generated, the differentiation from pre-Tfh (Tigit⁺) to GC-Tfh cells is severely blocked (Fig. 6).

Thus, our study suggests that there is a divergence of PD-1⁺CXCR5⁺CD4⁺ T cell differentiation: Tigit⁻PD-1⁺CXCR5⁺CD4⁺ T cells upregulate IL-7R α to give rise to CXCR5⁺CD4⁺ memory T cells, while Tigit⁺PD-1⁺CXCR5⁺CD4⁺ T cells differentiate further into GC-Tfh cells, and furthermore, in this pre- to GC-Tfh differentiation process, the factors c-Maf and Plekho1 play important roles.

2. Next, there are two major questions in the Tigit part. First, Tigit was upregulated in all antigen-specific cells at day 7 (early phase) in several models and then was downregulated on some cells at late time points. The central message claimed by the authors -- “that the sustained Tigit expression in PD-1⁺CXCR5⁺CD4⁺ T cells marked the precursor Tfh (pre-Tfh) to GC-Tfh transition” (abstract). This is not entirely supported by the data. Although the authors didn’t show, there were mixed non-Tfh, Tfh and GC-Tfh at day 7 in these models. At least at day 7, Tigit expression didn’t distinguish Tfh vs GC-Tfh and didn’t mark pre-Tfh to GC-Tfh transition.

Response: We thank the Reviewer for pointing out this important question which we haven’t explained well. In the influenza virus infection model, we found that there was no formation of germinal centers at day 6-7 p.i. (**new Fig. 1b**). Using a novel Bcl6-protein reporter mouse line, we found that at day 6-7 p.i., although there were some cells expressing high levels of PD-1 and CXCR5 (**new Fig. 1a**), the Bcl6 expression levels in such CD4⁺ T cells resembled those of pre-Tfh cells and were far lower than those of GC-Tfh cells (**new Fig. 1c**). Meanwhile, our data also showed that the CD4⁺ T cells at day 6-7 were in an extensive cell proliferation phase (Supplementary Fig. 1f). Thus, at this early stage of the immune response, it is mainly the CXCR5⁻ versus CXCR5⁺ divergence, and there are no GC-Tfh cells yet.

By day 14, when the extensive cell proliferation phase had passed (Supplementary Fig. 1f), the sustained Tigit expression marked only a subset of PD-1⁺CXCR5⁺CD4⁺ T cells. The combined results of Figs. 1-4 and 6 strongly suggest that Tigit⁺PD-1⁺CXCR5⁺CD4⁺ T cells have differentiated further into the GC-Tfh direction when compared to their Tigit⁻ counterparts. This leads us to conclude that if a PD-1⁺CXCR5⁺CD4⁺ T cell is on its way to become a GC-Tfh cell, the Tigit expression will be sustained. Now in the revised discussion, with the Reviewer’s comment in mind, we have added and explained that it is not the Tigit induction during the early stage of the response but rather the sustained expression after the early cell activation and proliferation phase is passed that marks the pre-Tfh to GC-Tfh differentiation.

3. There is at least another possibility for the different Tigit expression between Tfh vs GC-Tfh at late time points. GC-Tfh might exit GC and become memory cells with the downregulation of Tigit.

Response: We thank the Reviewer for bringing up this important point which we also discussed in our manuscript. CXCR5⁺CD4⁺ T cell memory versus GC-Tfh memory is certainly a long-standing question in the Tfh field. We have generated a novel GC-Tfh fate-mapping mouse line (data not shown). Our preliminary data show that at day 14, within the PD-1⁺CXCR5⁺CD4⁺ T cell population, the percentage of

PD-1⁺CXCR5⁺CD4⁺ T cells derived from GC-Tfh cells is low (data not shown). Thus, although at this stage about half of the PD-1⁺CXCR5⁺CD4⁺ T cells have downregulated Tigit, most of the Tigit⁻PD-1⁺CXCR5⁺CD4⁺ T cells are not GC-Tfh derived. At present we are conducting experiments to define GC-Tfh cell memory cells and their functions, and we hope the Reviewer would agree with us that these new results will be for future studies.

4. Several previous studies indicate that pre-Tfh to GC-Tfh transition and the divergence of differentiation happen early in the response, usually before day 7 (Kerfoot SM 2011; Choi YS 2011; He J 2013). It is not very suitable to address the question by focusing on 2 or 3 weeks after immunization or infection.

Response: In our response to question #2 of this Reviewer, we showed that by day 6-7 in the influenza virus infection, there was no GC formation or CD4⁺ T cells expressing GC-Tfh levels of Bcl6 (**new Fig. 1a-c**). We agree that the divergence differentiation could have happened a few days earlier than day 14, but we hope the Reviewer would agree with us that the approach in which we used the ‘snapshot’ data of day 14 or day 21 to investigate and understand the pre- to GC-Tfh cell differentiation program itself is valid. Consistently, the subsequent analyses and follow-up experiments using this approach have revealed new insight in our understanding of the divergence and differentiation and the pre-Tfh to GC-Tfh transition (Figs. 3-7).

5. Second, authors compared Tigit⁺ vs Tigit⁻ CXCR5⁺ population. As Tigit expression and CXCR5 expression are positively correlated (Fig 1F), I assume Tigit⁻ CXCR5⁺ population expresses high CXCR5 than Tigit⁻ CXCR5⁺ population. Therefore, the former is a partial GC-Tfh cells. The RNA-seq and ATAC-seq results were largely expected.

Response: We thank the Reviewer for agreeing with us that Tigit⁺CXCR5⁺ cells within the PD-1⁺CXCR5⁺CD4⁺ T cells (partial or early GC-Tfh cells) have differentiated further into the GC-Tfh direction than their Tigit⁻ counterparts.

As we mentioned in the introduction of the manuscript, although transcriptome and chromatin accessibility studies have been carried out to understand the Tfh cell differentiation, the majority of the studies compare CXCR5⁻ non-Tfh cells to bulk CXCR5⁺ Tfh cells and few have involved Bcl6^{hi} cells. The in-depth analysis of pre-Tfh to GC-Tfh differentiation is lacking, and little is known about the underlying mechanisms. In our study, we have carried out extensive analyses of the high-quality RNA-seq and ATAC-seq datasets and made the discoveries of novel factors that play important and stage-specific roles in GC-Tfh cell differentiation. We hope the Reviewer would agree with us that this is novel and significant.

6. Last, c-Maf’s function in both mouse and human Tfh generation and function have been extensively studied and reported. The two candidate targets of c-Maf-mediated regulation, Plekho1 and Cxxc5 are potentially interesting, but the characterization was superficial. Besides changing GC-Tfh percentages, it would be necessary to know how they might regulate T cell proliferation, survival or other T effector differentiation, as well as any Tfh functional alternation.

Response: We thank the Reviewer for these questions. Although c-Maf has been shown to be important for ‘Tfh cell differentiation’ in general, in our study, we have provided the new data showing that in the absence of c-Maf, while CXCR5⁺CD4⁺ T cells were still generated, there seemed to be a severe blockade of pre- to GC-Tfh cell development. More importantly, after extensive cross-examination of sequencing datasets, we have also identified novel c-Maf downstream factor(s) which implement a stage-specific regulation of GC-Tfh cell differentiation. We hope the Reviewer would agree with us that this is novel.

Plekho1 functions as a signaling molecule and Cxxc5 functions as a transcription factor and epigenetic regulator. To elucidate the molecular mechanisms underlying two novel factors in Tfh cell

differentiation simultaneously is challenging. Thus, we have decided to focus on one of them, *Plekho1*, and have revised the paper accordingly.

Using both shRNAmir and CRISPR/Cas9 approaches, we found that the manipulation of *Plekho1* levels did not affect non-Tfh cell differentiation in either competitive or non-competitive environments (Fig. 7d and **new Supplementary Fig. 7a, b**). In the non-competitive environment of the separate transfer into *Bcl6^{f/f}*CD4-Cre recipient mice, we also found that *Plekho1*-CRISPR GC-Tfh cell differentiation appeared to be normal, and they supported seemingly normal GC B cell differentiation, isotype switch, and LZ/DZ ratio (**new Supplementary Fig. 7b-e**). In the co-transfer experiments in a competitive environment, however, *Plekho1* becomes important for the competitive fitness of GC-Tfh cells (Fig. 7d and **new Supplementary Fig. 7a**).

To understand the mechanism underlying *Plekho1*-mediated regulation of the competitive fitness of GC-Tfh cells, we performed the new RNA-sequencing. The transcriptome analysis and Enrichr enrichment analysis (**new Fig. 7e-g** and **Supplementary Fig. 8**) suggested that the increased *Plekho1* expression in GC-Tfh cells would lead to more *Foxo1* activation (and translocation out of nucleus), which has been shown to promote GC-Tfh cell differentiation. Indeed, using ImageStream, we demonstrated that in sh*Plekho1* GC-Tfh cells, while total *Foxo1* expression levels were unchanged, there was increased retention of *Foxo1* in the nucleus (**new Fig. 7h**). Thus, supported by the new evidence, our study now reveals that during pre- to GC-Tfh transition, the increased *c-Maf* promotes *Plekho1* expression, which in turn leads to *Foxo1* activation and translocation out of nucleus, and this *c-Maf/Plekho1/Foxo1* axis plays a critical role in regulating the competitive fitness of GC-Tfh cells.

We want to thank all three Reviewers and the Editor for the constructive and helpful comments and suggestions, which led us to add critical new experiments that have made the discoveries more solid and exciting.

Rebuttal Letter Figures

Rebuttal Letter Fig. 1 | Tigit and IL-7Ra expression on c-Maf overexpressed PD-1⁺CXCR5⁺ donor cells. Purified OT-II cells from CD45.1⁺CD45.2⁺ and CD45.2⁺ OT-II^{Tg} mice were activated *in vitro* and infected with control retrovirus (RV-Ctrl) or retrovirus expressing *Maf* (RV-c-Maf), respectively. Retrovirally infected OT-II cells were co-transferred into CD45.1⁺ SMARTA recipient mice followed by intranasal infection with PR8-OVA two days after cell transfer. At days 14 (**a, b**) and 21 (**c, d**) p.i., donor OT-II cells in the medLN were analyzed for PD-1 and CXCR5 staining (**a, c**), and PD-1⁺CXCR5⁺ donor cells were analyzed for Tigit and IL-7Ra staining (**b, d**) (day 14, n = 5; day 21, n = 9). Data in (**a-d**) are representative (or pooled) results of two independent experiments. The *P*-values were determined by a two-tailed paired *t*-test (**a-d**).

Rebuttal Letter Fig. 2 | Tigit and IL-7R α expression on shPlekho1 PD-1⁺CXCR5⁺ donor cells. Purified OT-II cells from CD45.1⁺CD45.2⁺ and CD45.2⁺ OT-II^{Tg} mice were activated *in vitro* and infected with retrovirus expressing shRNAmir against *CD19* (shCD19) or *Plekho1* (shPlekho1), respectively. Retrovirally infected OT-II cells were co-transferred into CD45.1⁺ SMARTA recipient mice followed by intranasal infection with PR8-OVA two days after cell transfer. At days 14 (**a**) and 21 (**b**) p.i., the PD-1⁺CXCR5⁺ donor cells in the medLN were analyzed for Tigit and IL-7R α staining (day 14, n = 7; day 21, n = 5). Data in (**a**, **b**) are representative (or pooled) results of two independent experiments. The *P*-values were determined by a two-tailed paired *t*-test (**a**, **b**).

Rebuttal Letter Fig. 3 | Human PBMCs were analyzed by flow cytometry for Tigit, IL-7R α , and c-Maf expression. a Gating strategy for granulocytes, naïve CD4⁺CD45RA^{hi} T cells, Eff/Mem CD4⁺CD45RA^{lo} T cells, and the CXCR5 expression on Eff/Mem CD4⁺ T cells. **b** CD4⁺CD45RA^{hi}, CD4⁺CD45RA^{lo}CXCR5⁻, and CD4⁺CD45RA^{lo}CXCR5⁺ cells were analyzed for Tigit and IL-7R α staining. **c** Granulocytes, naïve CD4⁺CD45RA^{hi}, CD4⁺CD45RA^{lo}CXCR5⁻, and CD4⁺CD45RA^{lo}CXCR5⁺ T cells were analyzed for c-Maf intracellular staining.

REVIEWERS' COMMENTS

Reviewer #1 (Remarks to the Author):

The concerns raised in my prior review have been addressed. Otherwise, my prior review stands.

Reviewer #2 (Remarks to the Author):

The authors properly addressed my concerns.

Reviewer #3 (Remarks to the Author):

In the revision, the authors provide substantially new data to strengthen the conclusion. It is essential that the new data in Fig. 1 show GC-Tfh and GC are not formed at day 7, which justifies the feasibility of the model to study pre-GC-Tfh to GC-Tfh differentiation. Such new data have addressed my major concerns.

The new mechanistic results revealing the regulatory axis by cMaf-Plekho1-Fox1 are intriguing and add depth to the study. However, it is a bit surprising that c-Maf overexpression didn't enhance GC-Tfh formation in the rebuttal Fig 1. It has been shown that c-Maf overexpression enhanced CXCR5 and other Tfh makers in human cells (Kroenke MA et al. JI, 2012). Such results should be included in the paper.

I have three additional suggestions to be considered.

First, the suggestion from reviewer 1 to image the localization of TIGIT+ vs TIGIT- cells is a good one. However, the images in Fig 2 show little difference in TIGIT expression between GC vs follicular Tfh cells. Such results couldn't distinguish GC vs non-GC differentiation of precursor cells. The examination of circulating OT-II cells as the memory pool could be stronger evidence to demonstrate the downregulation of TIGIT is associated with egress from LN, oppositely to the commitment to the GC-Tfh fate. Although the examination of blood Tfh cells was less studied in mouse models, I noticed an old paper that has done so (He J et al. Immunity, 2013). According to the model suggested by the authors, there should be few blood Tfh cells at day 7. At day 14, there should be more blood Tfh cells with a TIGIT- phenotype. This could be nicely supported by the human PBMC data shown in rebuttal Fig. 3. These results should be included in the paper.

Second, the functional investigation of TIGIT was moved from a major figure to Suppl. Fig. 2. In my opinion, such a negative result is important and should remain in the main figure.

Last, did the knockdown or knockout of PLekho1 affect the total number of OT-II cells?

NCOMMS-22-39526B: The divergence of PD-1⁺CXCR5⁺CD4⁺ T cell differentiation and stage-specific regulation of GC-Tfh cells

Reviewer #3 (Remarks to the Author):

1. The new mechanistic results revealing the regulatory axis by cMaf-Plekho1-Foxo1 are intriguing and add depth to the study. However, it is a bit surprising that c-Maf overexpression didn't enhance GC-Tfh formation in the rebuttal Fig 1. It has been shown that c-Maf overexpression enhanced CXCR5 and other Tfh makers in human cells (Kroenke MA et al. JI, 2012). Such results should be included in the paper.

Response: Compared to c-Maf overexpression enhancing CXCR5 and a few Tfh markers (but not Bcl6) in human CD4⁺ T cells *in vitro*, we think that our results of c-Maf overexpression not enhancing GC-Tfh formation *in vivo* could be due to already significantly increased protein levels of endogenous c-Maf from PD-1⁺CXCR5⁺ pre-Tfh cells to PD-1^{hi}CXCR5^{hi} GC-Tfh cells (**Fig. 6f**), or it is possible that some other limiting factors (e.g. signaling upstream of Bcl6 protein expression) are more important. It is interesting that the overexpression of c-Maf also did not affect Tigit or IL-7R α expression in PD-1⁺CXCR5⁺CD4⁺ T cells (first review **Rebuttal Letter Fig. 1** in response to Question 3 of Reviewer #1), suggesting that the determination of Tigit⁺ versus Tigit⁻ PD-1⁺CXCR5⁺CD4⁺ T cells most likely does not involve c-Maf.

In **Figs. 6 and 7**, we focused on elucidating the stage-specific role of the c-Maf/Plekho1/Foxo1 axis in regulating GC-Tfh cells. As we have agreed to publish the reviewer comments to the authors and author rebuttal letters as a supplementary peer review file, the results of the c-Maf overexpression experiments included in the **Rebuttal Letter Fig.1** will be published as well.

2. The examination of circulating OT-II cells as the memory pool could be stronger evidence to demonstrate the downregulation of TIGIT is associated with egress from LN, oppositely to the commitment to the GC-Tfh fate. Although the examination of blood Tfh cells was less studied in mouse models, I noticed an old paper that has done so (He J et al. Immunity, 2013). According to the model suggested by the authors, there should be few blood Tfh cells at day 7. At day 14, there should be more blood Tfh cells with a TIGIT⁻ phenotype. This could be nicely supported by the human PBMC data shown in rebuttal Fig. 3. These results should be included in the paper.

Response: In **Fig. 4**, we have provided evidence demonstrating that it is the subset of Tigit⁻ cells within the PD-1⁺CXCR5⁺CD4⁺ T cell population in medLN and spleen that upregulate IL-7R α to become the memory precursor/memory cells. We think that "... the downregulation of Tigit is associated with egress from LN" could be a related but separate event happening to PD-1⁺CXCR5⁺CD4⁺ T cells, and the Reviewer's point is very interesting.

As raised in the first review **Reviewer#1 Question 4**, "The true nature of circulating TFH is a matter of great controversy". In addition to using a novel GC-Tfh fate-mapping reporter mouse line, we have also conducted a series of experiments to address the egress of CXCR5⁺CD4⁺ T cells into the blood in different models. The memory CXCR5⁺CD4⁺ T cells is a yet more complicated question. We would agree with the Reviewer's prediction that the Tigit expression on CXCR5⁺CD4⁺ T cells will be reduced from day 7 to day 14 in the blood, and we also hope the Reviewer would agree with us that this series of important questions regarding CXCR5⁺CD4⁺ T cell egress and memory in the blood needs a thorough and systematic evaluation for a separate study.

3. Second, the functional investigation of TIGIT was moved from a major figure to Suppl. Fig. 2. In my opinion, such a negative result is important and should remain in the main figure.

Response: We agree with the Reviewer that such a negative result is important. With the new results of Tigit immunofluorescence staining in **Fig. 2**, considering the figure title, flow of the text, and the space of

figure legend, we hope the Reviewer would agree with us that having the negative result of Tigit function in Suppl. Fig. 2 is not improper and still highlights its importance.

4. Last, did the knockdown or knockout of PLEkho1 affect the total number of OT-II cells?

Response: In both OT-II transfer into SMARTA recipient mice and into Bcl6^{f/f}CD4-Cre^{Tg} recipient mice, there was no difference in total number of OT-II T cells between the Plekho1-WT group and the Plekho1-CRISPR group.